# BaRC: Barrier Robustness Certificates for Neural Networks against Data Poisoning and Evasion Attacks

## Abstract

The increasing use of machine learning in safety-critical domains amplifies the risk of adversarial threats, especially *data poisoning attacks* that corrupt training data to degrade performance or induce unsafe behavior. Most existing defenses lack formal guarantees or rely on restrictive assumptions about the model class, attack type, extent of poisoning, or point-wise certification, limiting their practical reliability. This paper introduces a principled formal robustness certification framework that models gradient-based training as a *discrete-time dynamical system* (dt-DS) and formulates poisoning robustness as a formal safety verification problem. By adapting the concept of *barrier certificates* (BCs) from control theory, we introduce sufficient conditions to certify a robust radius within which the model's parameter trajectories during training remain safe under worst-case $\ell_p$-norm based poisoning. To make this practical, we parameterize BCs as neural networks trained on finite sets of poisoned trajectories. We further derive *probably approximately correct* (PAC) bounds by solving a *scenario convex program* (SCP), which yields a confidence lower bound on the certified robustness radius generalizing beyond the training set. Importantly, our framework also extends to certification against test-time attacks, making it the *first* unified framework to provide formal guarantees in both training and test-time attack settings. Experiments on MNIST, SVHN, and CIFAR-10 show that our approach certifies non-trivial perturbation budgets while being model-agnostic and requiring no prior knowledge of the attack or contamination level.

## 1 Introduction

The deployment of machine learning (ML) models in safety-critical domains, such as autonomous driving and medical diagnostics, increases the risk of adversarial threats, especially **data poisoning attacks**. In such attacks, an adversary deliberately injects crafted perturbations into the *training dataset* to subvert the model's behavior, degrade performance, or break the safety requirements at test-time (Biggio et al., 2012; Shafahi et al., 2018; Koh & Liang, 2017). These attacks exploit the training pipeline, embedding backdoors or stealth vulnerabilities that can persist unnoticed and trigger failures in mission critical applications (Carlini et al., 2024; Schwarzschild et al., 2021). Although a variety of defenses have been proposed to mitigate data poisoning attacks, ranging from detecting and removing poisoned samples to modifying training strategies for robustness, these approaches are largely heuristic and remain vulnerable to sophisticated adaptive attacks (Goldblum et al., 2023; Koh et al., 2022; Shafahi et al., 2018; Huang et al., 2020). This highlights the need to develop **formal robustness certificates** that guarantee that the predictions of a model remain unchanged by poisoning.

A small but growing line of work explores such robustness certification under fixed-threat models and a certain allowed corruption budget for poisoning. Notable techniques include randomized smoothing (Weber et al., 2023), model ensembling (Levine & Feizi, 2021), parameter-space interval bounds via convex relaxation (Sosnin et al., 2025), and combining kernels and linear programming approaches for large-width networks (Sabanayagam et al., 2025; Gosch et al., 2025). However, these methods face three major limitations: $(i)$ **Threat model and budget assumptions**: Most works assume a fixed (un)bounded corruption budget, with *no* mechanism to compute the budget

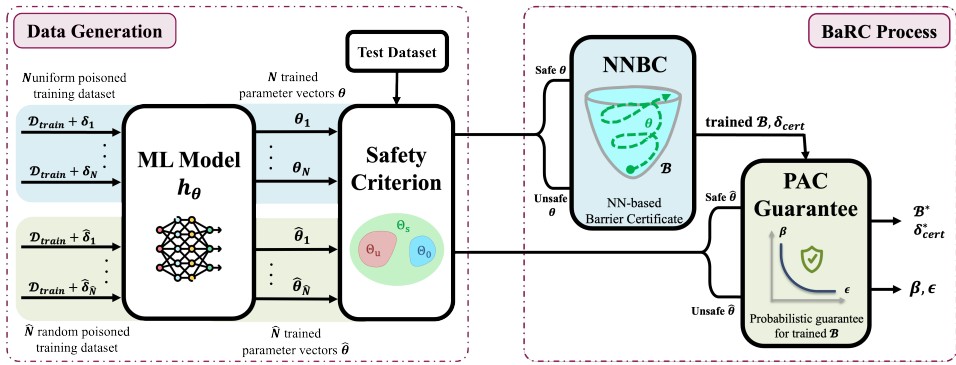

Figure 1: BaRC framework against data poisoning attacks. An NNBC $\mathcal{B}$ is learned from the parameters of $h_\theta$ and verified through a PAC bound guarantee, ensuring robustness with violation at most $\epsilon$ and a confidence of at least $1-\beta$.

corresponding to a desired robustness level. Furthermore, it is assumed that the number of corrupted data points is known (Weber et al., 2023); $(ii)$ **Model specificity**: The approaches are limited to specific architectures in some cases, such as decision trees (Meyer et al., 2021), nearest neighbors (Jia et al., 2022), or graph neural networks (Sabanayagam et al., 2025; Gosch et al., 2025), and assume white-box access; $(iii)$ **Pointwise guarantees**: Most certification methods provide guarantees only for individual test points, failing to account for the global behavior of the model in the test data (Levine & Feizi, 2021). Together, these limitations underscore a fundamental open question:
*Can we determine, for any ML model, a certified poisoning budget under $\ell_p$-norm based corruption such that the model's performance degradation under poisoning is at most a desired threshold $\alpha$?*

In this work, we answer this question positively by developing a framework inspired by *control-theoretic safety verification* to certify ML models against data poisoning. We model gradient-based training as a discrete-time dynamical system (dt-DS), where the model parameters form the system state and the (potentially poisoned) training data act as the input to the system. Within this dynamical system view, we recast poisoning robustness as a formal safety verification problem and adapt **barrier certificates (BCs)** (Ames et al., 2014; Prajna et al., 2007) to certify a robust radius $\ell_p$ for a prescribed accuracy-degradation tolerance $\alpha$, ensuring that parameter trajectories remain within the safe set under worst-case poisoning. This enables principled worst-case guarantees without requiring knowledge of the specific ML architecture, the poisoning attack strategy, or the fraction of corrupted data, and provides a certificate for the *entire* test dataset, not just point-wise test samples. We note that it is challenging to explicitly construct the BCs for ML training processes due to the high dimensionality of the parameter space, lack of a closed-form training model, and the unknown nature of the poisoning attack model, thus rendering the exact system dynamics inaccessible. To address this, we adopt a *data-driven* approach that parameterizes BC as a neural network, producing a **neural network-based BC (NNBC)**, similar to recent data-driven safety verification using BC for unknown systems (Anand & Zamani, 2023; Zhang et al., 2025; Rickard et al., 2025). Although the NNBC is trained on a finite set of trajectories generated under admissible perturbations, we ensure that the BC conditions hold more generally by reformulating verification as a *scenario convex program* (SCP). This allows us to derive **probably approximately correct (PAC)** bounds (Campi & Garatti, 2008; Rickard et al., 2025), providing a probabilistic guarantee. The PAC bound ensures, with some confidence, that the probability of violating the barrier conditions on unseen trajectories stays below a prescribed level, ensuring that the certified radius generalizes beyond the training trajectories. Figure 1 presents the BaRC framework against data poisoning attacks. Importantly, our NNBC framework allows for certifying test-time corruptions as well, providing a unified approach to certify both train and test data poisoning.

**Key novel contributions of this work are as follows:**
**1.** We cast gradient-based ML training as a discrete-time dynamical system and reformulate robustness certification against train and test data perturbations as a formal *safety verification* problem using *barrier certificates (BC)*.
**2.** We introduce a *scalable* neural network-based BC (NNBC) framework to overcome the intractability of the explicit BC design for high-dimensional and unknown poisoned training dynamics. NNBC is trained to obtain the certified robust radius, the largest admissible perturbation of the train or test data for which the degradation in test accuracy is provably at most a given threshold.

**3.** We derive a *probably approximately correct* (PAC) bound that provides a rigorous probabilistic guarantee for the trained NNBC and its associated certified robust radius.

**4.** Our approach is model-agnostic and does not require prior knowledge of the architecture, the poisoning attack strategy, or the amount of data corrupted, thus broadly applicable.

**5.** We empirically validate the effectiveness of our certification framework on various models and datasets, demonstrating its ability to quantify and formally certify safe perturbation budgets for training and test-time attacks in practice.

**Related Work.** Although determining the maximum allowable poisoning budget for a specified model performance is understudied, many different flavors of certificates are developed for data poisoning (see Appendix E for a detailed discussion). Ensemble-based certifications are generally developed assuming unbounded perturbations to the samples and provide robustness guarantees by aggregating over multiple base models trained on randomly subsampled datasets. These certificates establish a lower bound on the number of clean samples required to outweigh poisoned examples under majority voting (Levine & Feizi, 2021; Jia et al., 2021). Moreover, these methods typically assume the independence between base models. Weber et al. (2023) extends randomized smoothing, a test-time certification strategy, to poisoning by considering that a fixed pattern is injected into a subset of training and all test inputs and certifies the prediction of a smoothed classifier. Other methods assume a bounded adversary, where the perturbation budget and the number of poisoned examples are explicitly constrained. Gosch et al. (2025) takes a kernel-based approach that requires knowledge of the corrupted training data and the magnitude of the perturbations, providing guarantees via linear programming. Sosnin et al. (2025) introduces a gradient-based certification method based on convex relaxations and interval bounds, certifying robustness for convex losses trained with known corruption levels. However, these approaches rely on restrictive knowledge of the adversary's behavior and are limited to specific model families.

## 2 PRELIMINARIES

All proofs, expanded notation, algorithmic procedures, and additional experimental results are deferred to the Appendix.

### 2.1 SETUP FORMULATION

We consider a supervised learning problem defined on a clean training dataset $\mathcal{D}_{\text{train}} = \{(u_i, y_i)\}_{i=1}^n \subseteq \mathbb{R}^m \times \mathcal{Y}$, where $\mathcal{Y} := \{1, \ldots, k\}$, consisting of $n$ input–label pairs, where each feature vector $u_i \in \mathbb{R}^m$ is associated with a label $y_i \in \mathcal{Y}$. Similarly, let $\mathcal{D}_{\text{test}} = \{(u_i', y_i')\}_{i=1}^{n'} \subseteq \mathbb{R}^m \times \mathcal{Y}$ be a held-out test set of size $n'$. Let $h_\theta : \mathbb{R}^m \to \mathcal{Y}$ denote a parameterized ML model (e.g., a neural network) with parameter vector $\theta \in \mathbb{R}^d$. The model is trained by iteratively updating $\theta$ according to a gradient-based update rule $f : \mathbb{R}^d \times \mathcal{J} \to \mathbb{R}^d$, where $\mathcal{J}$ denotes auxiliary inputs (e.g., gradients or batch indices). The parameters are updated by:

$$\theta(t+1) = f(\theta(t), \mathcal{J}(t)) := \theta(t) - \gamma_t \nabla_\theta \mathcal{L}(h_{\theta(t)}, \mathcal{D}_{\text{train}}(t)), \tag{1}$$

where $\gamma_t > 0$ is the learning rate, $\mathcal{J}(t) := (\gamma_t, \mathcal{D}_{\text{train}}(t))$, and $\mathcal{D}_{\text{train}}(t) \subseteq \mathcal{D}_{\text{train}}$ denotes the complete data set or a batch sampled in iteration $t \in \mathbb{N}_0$. The model is trained on the dataset $\mathcal{D}_{\text{train}}$ by minimizing the empirical training loss $\mathcal{L}(h_\theta, \mathcal{D}_{\text{train}}) := \frac{1}{n} \sum_{i=1}^n \ell(h_\theta(u_i), y_i)$, where $\ell : \mathcal{Y} \times \mathcal{Y} \to \mathbb{R}^+$ is a non-negative pointwise loss function. Training is performed until convergence or a predefined termination criterion is met. The generalization performance of the trained model is evaluated on $\mathcal{D}_{\text{test}}$ via the test accuracy denoted by $g$, such that:

$$g(\theta) := \frac{1}{n'} \sum_{i=1}^{n'} \mathbf{1}_{\{h_\theta(u_i') = y_i'\}}. \tag{2}$$

In practice, the data used to train or evaluate an ML model $h_\theta$ may be adversarially perturbed, resulting in degraded performance. Such poisoning attacks can target input features, labels, or both, and may occur during either the training or testing phases of the ML pipeline. In this work, we focus on input-space poisoning, where perturbations affect the training or test data features. Our certification framework provides formal guarantees of the maximum allowable perturbation magnitude, measured in the $\ell_p$ norm. The following definitions formalize this poisoning threat model.

**Definition 1 (Train-time poisoning attack).** *Let $\mathcal{D}_{\text{train}} = \{(u_i, y_i)\}_{i=1}^n$ be the clean training set. A poisoning attack is modeled as an adversary $\mathcal{A}$ that perturbs an unknown fraction $\rho \in [0, 1]$ of the training samples, resulting in $r := \lceil \rho.n \rceil$ modified inputs. The poisoned dataset is given by:*

$$\mathcal{D}_{\text{train}}^{\Delta} := \left\{ (u_i + \delta_i, y_i) \right\}_{i=1}^r \cup \left\{ (u_i, y_i) \right\}_{i=r+1}^n, \quad s.t. \quad \|\Delta\|_p := \max_{i \in [r]} \|\delta_i\|_p \leq \delta, \quad (3)$$

*where $\Delta := [\delta_1, \ldots, \delta_r] \in \mathbb{R}^{r \times m}$ is the perturbation matrix, and each row $\delta_i \in \mathbb{R}^m$ perturbs the feature vector $u_i$ of the $i$-th training sample and is constrained by a row-wise $\ell_p$ norm bound.*

**Definition 2 (Test-time evasion attack).** *Let $\mathcal{D}_{\text{test}} = \{(u_i', y_i')\}_{i=1}^{n'}$ be the clean test set. An evasion attack is modeled as an adversary $\mathcal{A}'$ that perturbs an unknown fraction $\rho' \in [0, 1]$ of the test samples, resulting in $r' := \lceil \rho'.n' \rceil$ modified inputs. The perturbed test set is given by:*

$$\mathcal{D}_{\text{test}}^{\Delta'} := \left\{ (u_i' + \delta_i', y_i') \right\}_{i=1}^{r'} \cup \left\{ (u_i', y_i') \right\}_{i=r'+1}^{n'}, \quad s.t. \quad \|\Delta'\|_p := \max_{i \in [r']} \|\delta_i'\|_p \leq \delta', \quad (4)$$

*where $\Delta' := [\delta_1', \ldots, \delta_{r'}'] \in \mathbb{R}^{r' \times m}$ is the matrix of input-space perturbations applied at test-time.*

Note that, without loss of generality, we assume that the first $r$ elements of the training dataset and the first $r'$ elements of the test dataset are poisoned. In addition, if $\delta = 0$ or $\rho = 0$ (resp. $\delta' = 0$ or $\rho' = 0$), the poisoned dataset reduces to the clean one, i.e., $\mathcal{D}_{\text{train}}^{\Delta} = \mathcal{D}_{\text{train}}$ (resp. $\mathcal{D}_{\text{test}}^{\Delta'} = \mathcal{D}_{\text{test}}$).

## 2.2 METHODOLOGY

The goal of this paper is to formally certify the robustness of an ML model $h_\theta$ against data poisoning or evasion attacks by determining its certified robust radius under $\ell_p$ perturbations, within which the performance of the trained model remains above a target threshold. We now formalize this robustness certification problem:

**Problem 3 (Certified robust radius).** *Let $h_\theta$ be a parameterized ML model trained on a potentially poisoned training set $\mathcal{D}_{\text{train}}^{\Delta}$ and evaluated on a potentially poisoned test set $\mathcal{D}_{\text{test}}^{\Delta'}$. Given a threshold $\alpha \in [0, 1]$, the objective is to determine the largest poisoning radius $\delta_{\text{cert}}$ for training-time (resp. $\delta_{\text{cert}}'$ for test-time), such that, for all perturbations $\Delta$ (resp. $\Delta'$) satisfying $\|\Delta\|_p \leq \delta_{\text{cert}}$ (resp. $\|\Delta'\|_p \leq \delta_{\text{cert}}'$), the performance degradation of the trained model remains within $\alpha$.*

During training, data poisoning can alter the optimization process, causing convergence to suboptimal or unsafe regions in the parameter space. At test-time, evasion attacks can shift the decision boundary, degrading generalization and reliability. We capture these effects through a safety criterion that distinguishes safe from unsafe parameter regions with respect to a given gap threshold.

**Definition 4 (Safety criterion).** *Let $h_\theta$ be an ML model with parameters $\theta(t) \in \mathbb{R}^d$ at iteration $t \in \mathbb{N}_0$, initialized at $\theta(0) \in \Theta_0$, trained on a possibly poisoned dataset $\mathcal{D}_{\text{train}}^{\Delta}$, and converges at $t = t_\infty$. To quantify degradation in test performance under poisoning, we define the safety criterion as the accuracy drop of $h_\theta$ relative to clean training: $\mathcal{G}(\theta(t')) = g_c(\theta(t')) - g_p(\theta(t'))$, where $g_c(\theta(t'))$ denotes the test accuracy of the model trained on the clean dataset, and $g_p(\theta(t'))$ denotes the test accuracy of the model trained on the poisoned dataset; both accuracies are evaluated at iteration $t = t'$. Given a threshold $\alpha \in [0, 1]$, we define the safe and unsafe sets, as follows:*

$$\Theta_s := \left\{ \theta \in \mathbb{R}^d \mid \mathcal{G}(\theta) \leq \alpha \right\}, \qquad \Theta_u := \left\{ \theta \in \mathbb{R}^d \mid \mathcal{G}(\theta) > \alpha \right\}. \quad (5)$$

*Consequently, we define the empirical train-time robust radius $\delta_{\text{emp}}$ (resp. test-time radius $\delta_{\text{emp}}'$) as the largest perturbation bound such that, for all $\Delta$ (resp. $\Delta'$) with $\|\Delta\|_p \leq \delta_{\text{emp}}$ (resp. $\|\Delta'\|_p \leq \delta_{\text{emp}}'$), the terminal parameters $\theta(t_\infty)$ satisfy $\mathcal{G}(\theta(t_\infty)) \leq \alpha$, and thus remain within the safe set $\Theta_s$.*

**Remark 5.** *Note that Definition 4 is generally stated to allow for reasoning and evaluating $\mathcal{G}(\theta(t'))$ at any iteration. However, the robustness objective in Problem 3 and our focus in this work is to certify that the terminal parameters $\theta(t_\infty)$ are in the safe set, i.e., $\mathcal{G}(\theta(t_\infty)) \leq \alpha$.*

Although empirical robust radii estimate robustness on poisoned data, they lack formal guarantees for all trajectories within the admissible budget realizations. To bridge this critical gap, we model the training process of $h_\theta$ as a dt-DS, enabling the derivation of conditions to certify a robust radius.

**Definition 6 (Dynamical system).** *Let $h_\theta$ be an ML model. A discrete-time dynamical system (dt-DS) is a tuple $\mathfrak{S} = (\Theta, \Theta_0, \mathcal{D}^\Delta_{\text{train}}, f)$, where $\mathcal{D}^\Delta_{\text{train}} \subseteq \mathbb{R}^m \times \mathcal{Y}$ is a (potentially poisoned) training dataset, $\Theta \subseteq \mathbb{R}^d$ is the set of model parameters, $\Theta_0 \subseteq \Theta$ is the set of initial model parameters, and $f : \mathbb{R}^d \times \mathbb{R}^m \times \mathcal{Y} \to \mathbb{R}^d$ is the parameter update map specified by the gradient-based optimization algorithm in Section 2.1. Thus, the system $\mathfrak{S}$ evolves according to equation (1), i.e., $\theta(t + 1) = f\big(\theta(t), \mathcal{J}(t)\big)$, for all $t \in \mathbb{N}_0$, where $\theta(t) \in \Theta$ and $\mathcal{J}(t) = (\gamma_t, \mathcal{D}^\Delta_{\text{train}}(t))$ represent the state and input of the system at time $t$, respectively.*

Since the learning rate $\gamma_t$ is treated as a fixed hyperparameter, we omit it from the update rule notation in the rest of the paper and write $f(\theta, \mathcal{D}^\Delta_{\text{train}})$ for notational simplicity. We then adopt a BC formulation, inspired by Prajna & Jadbabaie (2004), as the foundation of our robustness certification.

**Definition 7 (Barrier certificate).** *Let $h_\theta$ be an ML model with its associated dt-DS $\mathfrak{S} = (\Theta, \Theta_0, \mathcal{D}^\Delta_{\text{train}}, f)$. Consider $\mathcal{G}$ be the safety criterion function as in definition 4. Let $\delta_{\text{emp}}$ (resp. $\delta'_{\text{emp}}$) be the empirical train-time (resp. test-time) robust radius, and let $\Theta_u \subseteq \Theta$ be the corresponding unsafe set, derived by training $h_\theta$ on $\mathcal{D}^\Delta_{\text{train}}$ and evaluating performance via g on $\mathcal{D}^{\Delta'}_{\text{test}}$. A function $\mathcal{B} : \mathbb{R}^d \to \mathbb{R}$ is called a barrier certificate (BC) if there exists a certified train-time robust radius $\delta_{\text{cert}}$ (resp. test-time radius $\delta'_{\text{cert}}$) such that the following conditions hold for all (potential) perturbations satisfying $\|\Delta\|_p \leq \delta_{\text{cert}}$ (resp. $\|\Delta'\|_p \leq \delta'_{\text{cert}}$):*

$$\mathcal{B}(\theta) \leq 0, \qquad\qquad\qquad \forall \theta \in \Theta_0, \tag{6}$$

$$\mathcal{B}(\theta) > 0, \qquad\qquad\qquad \forall \theta \in \Theta_u, \tag{7}$$

$$\mathcal{B}(f(\theta, \mathcal{D}^\Delta_{\text{train}})) \leq 0, \qquad\qquad \forall \theta \in \Theta, \text{ s.t. } \mathcal{B}(\theta) \leq 0. \tag{8}$$

**Remark 8.** *Train-time poisoning alters the dynamics $f(\theta, \mathcal{D}^\Delta_{\text{train}})$ and affects the reachability condition (8), while test-time poisoning modifies only $\mathcal{D}^{\Delta'}_{\text{test}}$, influencing the output of the safety function $g(\theta)$. However, the form of the BC conditions remains identical for both cases, since $\mathcal{B}$ certifies safety based solely on terminal parameters, without requiring knowledge of the source of the perturbation.*

**Remark 9.** *Conditions (6) and (7) ensure that every admissible initialization $\theta(0) \in \Theta_0$ lies within the barrier sublevel set. Note that a safe initial parameter does not guarantee that the robustness certificate holds for the ML model. As detailed in Section 2.1, the model must still be trained to attain the desired functionality, during which it may become unsafe at convergence. For this reason, we enforce condition (8) to ensure the safety of the final trained model.*

The following theorem shows that the existence of a BC $\mathcal{B}$ implies a certified robust radius for the ML model. For more details about the role of $\mathcal{B}$ and the proof, see Appendix B.1.

**Theorem 10 (Certified robust radius).** *Let $h_\theta$ be an ML model, trained on a (potentially) poisoned dataset $\mathcal{D}^\Delta_{\text{train}}$ and evaluated on a (potentially) poisoned dataset $\mathcal{D}^{\Delta'}_{\text{test}}$, as in Section 2.1. Consider a dt-DS $\mathfrak{S} = (\Theta, \Theta_0, \mathcal{D}^\Delta_{\text{train}}, f)$ as in Definition 6, modeling the training dynamics of $h_\theta$. Let $\delta_{\text{emp}}$ (resp. $\delta'_{\text{emp}}$) denote the empirically derived train-time (resp. test-time) robust radius, and let $\Theta_s \subseteq \Theta$ and $\Theta_u \subseteq \Theta$ denote the corresponding safe and unsafe sets of terminal parameters, as introduced in Definition 4. If there exists a BC $\mathcal{B}$ satisfying the conditions in Definition 7 for a train-time (resp. test-time) robust radius $\delta_{\text{cert}}$ (resp. $\delta'_{\text{cert}}$), then all trajectories initialized at $\theta(0) \in \Theta_0$ remain within the safe set $\Theta_s$ and never enter the unsafe set $\Theta_u$ for any perturbations $\|\Delta\|_p \leq \delta_{\text{cert}}$ (resp. $\|\Delta'\|_p \leq \delta'_{\text{cert}}$). Thus, $\delta_{\text{cert}}$ (resp. $\delta'_{\text{cert}}$) serves as a certified train-time (resp. test-time) robust radius, ensuring that, under worst-case perturbations, the degradation in test accuracy is at most $\alpha$.*

However, constructing a BC $\mathcal{B}$ for $\mathfrak{S}$ is intractable due to the high dimensionality of the model parameters and the lack of an explicit mathematical model of $\mathfrak{S}$. To overcome this, we propose a data-driven approach to synthesize $\mathcal{B}$.

## 3 DATA-DRIVEN ROBUSTNESS CERTIFICATION

In this section, we present our data-driven approach to certify the robustness of an ML model $h_\theta$ by constructing a neural network-based barrier certificate (NNBC) whose parameters are trained to satisfy the conditions in Definition 7. Given a dynamical system $\mathfrak{S}$, we define an NNBC $\mathcal{B}_\varphi : \mathbb{R}^d \to \mathbb{R}$ parameterized by $\varphi$. The input to the network is the state vector $\theta \in \mathbb{R}^d$, and the output is a scalar barrier value $\mathcal{B}_\varphi(\theta)$. The network uses ReLU activations in the hidden layers to ensure a piecewise

affine, locally Lipschitz structure, and an identity activation in the output layer. The depth and width of $\mathcal{B}_\varphi(\theta)$ are tunable hyperparameters.

**Data.** To train an NNBC $\mathcal{B}_\varphi$, we generate a dataset by training $h_\theta$ under varying levels of data poisoning across $N$ uniformly spaced budgets. Specifically, we define two grids $\delta_{\text{grid}} = \langle \delta_1, \ldots, \delta_N \rangle$, and $\delta'_{\text{grid}} = \langle \delta'_1, \ldots, \delta'_N \rangle$, where each point in $\delta_{\text{grid}}$ and $\delta'_{\text{grid}}$, represent train- and test-time poisoning levels, respectively. Depending on the certification type (train- or test-time), one grid is fixed to zero. Then, for each $i \in \{1, \ldots, N\}$, we randomly initialize $h_\theta$ at $\theta_i(0) \in \Theta_0$ and train it on $\mathcal{D}_{\text{train}}^\Delta$ with $\|\Delta\|_p = \delta_i$, yielding the terminal state $\theta_i(t_\infty)$. Next, following Definition 4, each trained model is evaluated on the dataset $\mathcal{D}_{\text{test}}^{\Delta'}$, where $\|\Delta'\|_p = \delta'_i$, and classified as safe or unsafe based on the safety criterion $\mathcal{G}(\theta_i(t_\infty))$ compared to a threshold $\alpha \in [0, 1]$. Thus, the resulting datasets are:

$$\mathcal{I} = \big\{ \theta_i(0) \, | \, \theta_i(0) \in \Theta_0 \big\}, \quad \mathcal{S} = \big\{ \theta_i(t_\infty) \, | \, \mathcal{G}(\theta(t_\infty)) \leq \alpha \big\}, \quad \mathcal{U} = \big\{ \theta_i(t_\infty) \, | \, \mathcal{G}(\theta(t_\infty)) > \alpha \big\}, \quad (9)$$

where $\mathcal{I} \subseteq \Theta_0$, $\mathcal{S} \subseteq \Theta_s$, and $\mathcal{U} \subseteq \Theta_u$, and we denote the union $\vartheta := \mathcal{I} \cup \mathcal{S} \cup \mathcal{U} \subseteq \Theta$. Then $\delta_{\text{emp}}$ or $\delta'_{\text{emp}}$ is computed. Note that, depending on the chosen threshold $\alpha$, some sampled sets may be empty; we refer interested readers to the Appendix for more details.

**Loss.** In order to train an NNBC $\mathcal{B}_\varphi$ that satisfies the conditions in Definition 7 across the generated datasets in (9), we define a composite loss function, as follows for all $i \in \{1, \ldots, N\}$:

$$\mathcal{L}(\varphi) = c_\mathcal{I} \sum_{\theta_i \in \mathcal{I}} \mathcal{R}\big(\mathcal{B}_\varphi(\theta_i)\big) + c_\mathcal{U} \sum_{\theta_i \in \mathcal{U}} \mathcal{R}\big(-\mathcal{B}_\varphi(\theta_i)\big) + c_\mathcal{Z} \sum_{\theta_i \in \mathcal{Z}} \mathcal{R}\big(\mathcal{B}_\varphi\big(f(\theta_i, \mathcal{D}_{\text{train}}^{\Delta_i})\big)\big), \quad (10)$$

where $\mathcal{R} := \text{ReLU}$ and $\|\Delta_i\|_p \leq \delta_{\text{cert}}$. The scalars $c_\mathcal{I}, c_\mathcal{U}, c_\mathcal{Z} > 0$ weight the respective conditions, and the set $\mathcal{Z} \subseteq \vartheta$, is defined by $\mathcal{Z} = \{\theta_i \mid \theta_i \in \vartheta, \ \mathcal{B}_\varphi(\theta_i) \leq 0\}$. When the loss function satisfies $\mathcal{L}(\varphi) = 0$, the NNBC $\mathcal{B}_\varphi$ is considered trained and denoted by $\mathcal{B}_\varphi^*$, along with the certified robust radius denoted by $\delta_{\text{cert}}^*$ (resp. $\delta_{\text{cert}}'^*$). This implies that the conditions (6)–(8) hold for all sampled states $\mathcal{I}, \mathcal{U}$, and $\mathcal{S}$. However, since NNBC $\mathcal{B}_\varphi^*$ is trained based on a finite set of parameter states, it does not cover the entire set $\Theta$. To overcome this limitation, we establish a probabilistic guarantee that extends the validity of the certificate beyond the training samples with some confidence.

## 4 PROBABILISTIC ROBUSTNESS GUARANTEE FOR NNBC

We now provide a formal robustness guarantee that quantifies how well the certified robust radius generalizes beyond the training samples. Specifically, we derive a probably approximately correct (PAC) guarantee with explicit confidence. To do this, we first need to assume that the learned NNBC $\mathcal{B}_\varphi^*$ is fixed and given to us. Let us define $\Theta_\mathcal{Z} = \{\theta \mid \theta \in \Theta, \mathcal{B}_\varphi(\theta) \leq 0\} \subseteq \Theta$. We then introduce a scalar margin denoted by $\eta_r \leq 0$ and functions $q_k : \mathbb{R}^d \to \mathbb{R}$, where $k \in \{1, 2, 3\}$, corresponding to the conditions in Definition 7, such that:

$$q_1(\theta, \eta_r) = \big(\mathcal{B}_\varphi^*(\theta) - \eta_r\big)\mathbf{1}_{\Theta_0}, \quad (11)$$

$$q_2(\theta, \eta_r) = \big(-\mathcal{B}_\varphi^*(\theta) - \eta_r\big)\mathbf{1}_{\Theta_u}, \quad (12)$$

$$q_3(\theta, \Delta, \eta_r) = \big(\mathcal{B}_\varphi^*(f(\theta, \mathcal{D}_{\text{train}}^\Delta)) - \eta_r\big)\mathbf{1}_{\Theta_\mathcal{Z}}. \quad (13)$$

**Robust convex problem (RCP).** To robustly verify the BC conditions (11)-(13) under all possible poisoning perturbations $\|\Delta\|_p \leq \delta_{\text{cert}}^*$ (resp. $\|\Delta'\|_p \leq \delta_{\text{cert}}'^*$), we formulate an RCP over the only decision variable $\eta_r$, enforcing strict satisfaction of all constraints:

$$\text{RCP} : \begin{cases} \min_{\eta_r \leq 0} & \eta_r \\ \text{s.t.} & q_k(\theta, \Delta, \eta_r) \leq 0, \quad \forall \theta \in \Theta, \forall k \in \{1, 2, 3\}. \end{cases} \quad (14)$$

Since $\mathcal{B}_\varphi^*$ is fixed, the RCP is a robust linear program over the scalar variable $\eta_r$, with the optimal value denoted by $\eta_r^*$. A solution $\eta_r^* < 0$ certifies that $\mathcal{B}_\varphi^*$ satisfies the conditions in Definition 7, and thus provides an exact robustness certificate for the poisoning attack with the corresponding radius $\delta_{\text{cert}}^*$ (resp. $\delta_{\text{cert}}'^*$) with a guarantee 100%. However, solving this robust linear program is intractable, as the state transition map $f$ is not available under unknown poisoning attacks and the

robust problem involves infinitely many constraints due to $\theta$ and $\Delta$ belonging to some continuous sets. To make this tractable, we relax the RCP into the following chance-constrained problem (CCP):

$$\text{CCP} : \begin{cases} \min_{\eta_r \leq 0} & \eta_r \\ \text{s.t.} & \mathbb{P}\left[q_k(\theta, \Delta, \eta_r) \leq 0, \ \ \forall \theta \in \Theta, \forall k \in \{1, 2, 3\}\right] \geq 1 - \epsilon, \end{cases} \tag{15}$$

where $\epsilon \in (0, 1)$ denotes the given violation probability. The goal is to solve the CCP in (15) rather than the RCP in (14). The CCP optimally discards a constraint subset of probability mass at most $\epsilon$ to maximize objective improvement. However, solving CCP is still challenging since both $\theta$ and $\Delta$ lie in continuous spaces. Therefore, we tackle the associated Scenario Convex Problem (SCP).

**Scenario convex problem (SCP).** We approximate infinitely many constraints by sampling $\hat{N}$ i.i.d. scenarios using the data generation process described in (9). This yields sampled sets $\mathcal{Z}_1 \subset \Theta_0$, $\mathcal{Z}_2 \subset \Theta_u$, and $\vartheta' \subset \Theta$ corresponding to data points in the initial, unsafe, and safe sets, respectively. Then, SCP enforces the inequalities only in these sampled scenarios for all $i \in \{1, \ldots, \hat{N}\}$ and $\|\Delta_i\|_p \leq \delta^*_{\text{cert}}$ (resp. $\|\Delta'_i\|_p \leq \delta'^*_{\text{cert}}$) as follows:

$$\text{SCP} : \begin{cases} \min_{\eta_s \leq 0} & \eta_s \\ \text{s.t.} & q_k(\theta_i, \Delta_i, \eta_s) \leq 0, \quad \forall \theta_i \in \mathcal{Z}_k, \forall k \in \{1, 2, 3\}. \end{cases} \tag{16}$$

where $\mathcal{Z}_3 = \{\theta_i \mid \theta_i \in \vartheta', \mathcal{B}_\varphi(\theta_i) \leq 0\}$ Let $\eta_s^*$ denote the optimal value of the SCP. Since the SCP replaces the infinite set with finitely many trajectories, it is crucial to assess the generalization of this solution. Hence, we establish a probabilistic bound that quantifies the gap between $\eta_s^*$ and $\eta_r^*$, guarantees the constructed BC $\mathcal{B}_\varphi^*$, and thus certifies the robust radii with some confidence.

**Probably approximately correct (PAC) guarantee.** To rigorously connect the CCP, and SCP, we adopt the PAC guarantee based on Theorem 1 of Calafiore & Campi (2006). Specifically, with a confidence of at least $1 - \beta$, the solution $\eta_s^*$ of SCP in (16) is a feasible solution of CCP in (15), provided that the number of i.i.d. scenarios $\hat{N}$ satisfies:

$$\hat{N} \geq \lceil \frac{\ln(\beta)}{\ln(1 - \epsilon)} \rceil. \tag{17}$$

We now present the main theoretical result of BaRC. A summary of the certification procedure is given in Algorithm 1, with extended versions for poisoning and evasion attacks in the Appendix C.

**Theorem 11 (BaRC).** *Let $h_\theta$ be an ML model, trained on a potentially poisoned training dataset $\mathcal{D}^\Delta_{\text{train}}$ and evaluated on a potentially poisoned test dataset $\mathcal{D}^{\Delta'}_{\text{test}}$, as described in Section 2.1. Assume that model updates follow the gradient-based rule $f$ as in (1), with the training process modeled by a dt-DS $\mathfrak{S}$. Consider $\alpha \in [0, 1]$, $\epsilon \in (0, 1)$, and $\beta \in [0, 1]$ as a gap threshold as in (5), the probability of violation as in (15), and the confidence level as in (17), respectively. Suppose an NNBC $\mathcal{B}_\varphi^*$ is trained using a finite number of samples generated through the procedure in Section 9, producing a certified train-time robust radius $\delta^*_{\text{cert}}$ (resp. certified test-time robust radius $\delta'^*_{\text{cert}}$). Let $\eta_s^* < 0$ be the optimal barrier margin obtained by solving the SCP in (16) on $\hat{N}$ i.i.d. samples, with $\hat{N}$ satisfying the bound in (17). Then, with a confidence of at least $1 - \beta$, the learned certificate $\mathcal{B}_\varphi^*$ ensures that, for all poisoning perturbations satisfying $\|\Delta\|_p \leq \delta^*_{\text{cert}}$ (resp. $\|\Delta'\|_p \leq \delta'^*_{\text{cert}}$), the model's converged parameters remain in the safe set $\Theta_s$ and $\mathcal{G}(\theta(t_\infty)) \leq \alpha$ as in Definition 4, with the violation probability of at most $\epsilon$.*

## 5 Experimental Results

In this section, we evaluate the effectiveness of BaRC and analyze how key design choices impact the certified robustness of the trained model $h_\theta$ under the $\ell_\infty$ and $\ell_2$ train-time threat models. Additional experiments for test-time certification are provided in the Appendix D. We perform experiments on three standard image classification benchmarks: **MNIST**, **SVHN**, and **CIFAR-10**. Robustness is assessed against three representative poisoning strategies during training, Projected Gradient Descent (PGD) (Madry et al., 2018), Backdoor Attack (BDA) (Gu et al., 2017), and Bullseye Polytope Attack (BPA) (Aghakhani et al., 2021), and against PGD and AutoAttack (AA) (Croce & Hein, 2020) at test time. The hypothesis class $h_\theta$ spans multiple architectures, including **MLP**, **CNN**, and **ResNet**, trained with optimizers **GD**, **SGD**, and **Adam**. We provide the details of the experimental setups with hyperparameters in Table 3, and list all model architectures in Table 4 in the Appendix.

---

**Algorithm 1** BaRC against Poisoning Attacks

---

**Input**: Model $h_\theta$, gap threshold $\alpha$, number of samples for training NNBC $N$, number of scenario for solving SCP $\hat{N}$, training horizon $t_\infty$, confidence level $1-\beta$, max iterations $T$

**Output**: Trained $\mathcal{B}_\varphi^*$, certified robust radius $\delta_{\text{cert}}^*$, violation probability $\epsilon$

1: Sample $N$ poisoned training trajectories dataset.
2: Train $h_\theta$ on each to obtain $\theta_i$ for all $i \in \{1, \ldots, N\}$. Collect terminal parameters $\theta_i(t_\infty)$ and label as safe or unsafe using $\mathcal{G}(\theta_i(t_\infty)) \leq \alpha$; assign $\theta_i(0)$ to initial set.
3: Fix $\delta_{\text{emp}} \leftarrow \max\{\delta_i \mid \mathcal{G}(\theta_i(t_\infty)) \leq \alpha\}$ and initialize $\delta_{\text{cert}} \leftarrow \delta_{\text{emp}}$.
4: Train an NNBC $\mathcal{B}_\varphi$ on collected data to satisfy $\mathcal{L} = 0$.
5: If there is no $\mathcal{B}_\varphi^*$, reduce $\delta_{\text{cert}}$ or increase $N$ and retrain. If not feasible after $T$ tries, $h_\theta$ can not be certified.
6: Generate $\hat{N}$ new i.i.d. poisoned samples using Step 1.
7: Solve the SCP in equation (16) to obtain the margin $\eta_s^*$. If $\eta_s^* > 0$, reduce $\delta_{\text{cert}}$ and return to Step 2 until $\eta_s^* \leq 0$. Then, compute the minimum violation probability $\epsilon$ from condition (17).

---

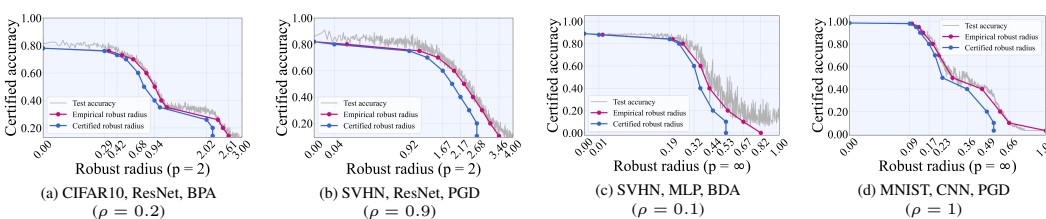

(a) CIFAR10, ResNet, BPA    (b) SVHN, ResNet, PGD    (c) SVHN, MLP, BDA    (d) MNIST, CNN, PGD
($\rho = 0.2$)         ($\rho = 0.9$)         ($\rho = 0.1$)         ($\rho = 1$)

Figure 2: Certified accuracy ($g_{\text{p}}^*$) versus perturbation magnitude ($\delta$) on different settings and poisoning scenarios. Each figure reports the terminal test accuracy $g(\theta(t_\infty))$, the empirical robust radius $\delta_{\text{emp}}$, and the certified robust radius $\delta_{\text{cert}}^*$ obtained using the proposed BaRC framework. The confidence level is fixed at $1 - \beta$, $\beta = 10^{-4}$, across all settings, with the corresponding violation probabilities being $\epsilon$ = (a) 0.015, (b) 0.013, (c) 0.006, and (d) 0.005.

**Results.** We present representative results of different combinations of $\ell_2$ and $\ell_\infty$ *train-time* attacks under poisoning ratio $\rho$ ranging from 0.1 to 1 and datasets, with a confidence level of $99.99\%$ in Table 1 and Figure 2 (see Table 3 and Figure 5 for all the results). We denote the *certified accuracy* by $g_{\text{p}}^*$ which is computed by $g_{\text{p}}^* = g_{\text{c}} - \alpha$ as in Definition 4. **Non-trivial certificates** are obtained in all settings. Exemplary, for SVHN, at $g_{\text{p}}^* = 0.75$ under PGD ($\ell_2$), BaRC certifies robust radii up to $\delta_{\text{cert}}^* = 0.92$ even when the poisoning ratio is as high as $\rho = 0.9$. All SCP margins $\eta_s^*$ are non-positive, ensuring feasibility, and the violation probability $\epsilon$ remains below 0.02 in most configurations (Table 1). Importantly, BaRC results in **tight certificates**, as the certified robust radius $\delta_{\text{cert}}^*$ is consistently close to the empirical robust radius $\delta_{\text{emp}}^*$.

Additionally, we compare BaRC with RAB, a randomized smoothing–based certified defense against evasion and backdoor attacks (Weber et al., 2023). As shown in Figure 3, *BaRC consistently achieves stronger and tighter guarantees than RAB*. Notably, BaRC more closely matches the empirical robustness (see Table 5 for more results). Other feature-poisoning certificates are less suitable for direct comparison: BagFlip (Zhang et al., 2022) is restricted to $\ell_0$ corruptions; ensemble-based methods permit unbounded perturbations (Levine & Feizi, 2021; Wang et al., 2022); and model-specific approaches tailored to neural networks either apply only to infinite-width graph neural networks (Gosch et al., 2025) or impose unrealistic restrictions on

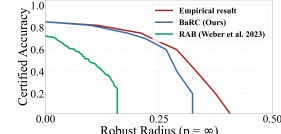

Figure 3: Comparison of BaRC and RAB on SVHN under test-time BDA with $\ell_\infty$ attack. BaRC consistently yields higher certified robustness than RAB.

training and model choice (Sosnin et al., 2025). More experiments, including diverse train- and test-time attack scenarios and a runtime analysis, are provided in Appendix D.

## 6   DISCUSSION AND CONCLUSION

Our proposed framework, BaRC, offers a principled, model-agnostic, attack-independent, data-driven solution for certifying both train-time and test-time poisoning of an ML model. We model gradient-based training as a dt-DS and frame poisoning robustness as a safety verification problem in parameter space. BaRC employs neural network–based barrier certificates, trained on sampled poisoning trajectories, and certifies robustness with probably approximately correct guarantees.

| Dataset | ML model | Optimizer | $\mathcal{A}$ | $\rho$-ratio | $p$-norm | N | $\hat{N}$ | $g_p^*$ | $\delta_{\text{emp}}$ | $\delta_{\text{cert}}^*$ | $\eta_s^*$ | $\epsilon$ |
|---------|----------|-----------|---------------|--------------|----------|---|-----------|---------|----------------------|--------------------------|-----------|-----------|
| MNIST | CNN | SGD | PGD | 1 | $\infty$ | 4000 | 1800 | 0.90 0.80 | 0.14 0.18 | 0.13 0.16 | -0.01 -0.01 | 0.005 |
| SVHN | ResNet | Adam | PGD | 0.9 | 2 | 3000 | 700 | 0.75 0.60 | 1.21 2.0 | 0.92 1.67 | -0.11 -0.05 | 0.013 |
| | MLP | SGD | BDA | 0.1 | $\infty$ | 4000 | 1500 | 0.80 0.60 | 0.25 0.35 | 0.23 0.32 | -0.02 -0.03 | 0.006 |
| CIFAR-10 | ResNet | Adam | BPA | 0.2 | 2 | 2000 | 600 | 0.70 0.60 | 0.61 0.84 | 0.52 0.68 | -0.01 -0.02 | 0.015 |

Table 1: Certification results for the train-time $\ell_2$ and $\ell_\infty$ poisoning attacks and datasets in Figure 2. Each row reports the certified accuracy ($g_p^*$), the empirical ($\delta_{\text{emp}}$) and certified ($\delta_{\text{cert}}^*$) radii, BC margin ($\eta_s^*$), and violation probability ($\epsilon$), evaluated at a performance gap threshold $\alpha$ and a confidence level of at least 99.99%. Larger $\delta_{\text{cert}}^*$ indicates stronger certified robustness.

**BaRC for train-time vs. test-time.** BaRC certifies robustness against both train-time and test-time poisoning, though these settings affect the learning differently. Train-time poisoning modifies the training data $\mathcal{D}_{\text{train}}^\Delta$, altering the learning dynamics and influencing which parameters are reachable, directly impacting the reachability-based constraints used in constructing the barrier certificate. In contrast, test-time poisoning affects only the evaluation data $\mathcal{D}_{\text{test}}^{\Delta'}$, modifying the safety predicate $g(\theta)$ by changing how the final model is judged as safe or unsafe. Despite these differences, the barrier certificate $\mathcal{B}_\varphi$ enforces the same structural constraints in both settings, and the certified radii ensure that the terminal model parameters remain within the safe set under each poisoning modality.

**Empirical robust radius vs. certified robust radius.** Any certified robust radius $\delta_{\text{cert}}$ (resp. $\delta_{\text{cert}}'$ for test-time) satisfies $\delta_{\text{cert}} \leq \delta_{\text{emp}}$ (resp. $\delta_{\text{cert}}' \leq \delta_{\text{emp}}'$). Ideally, a tight certificate would achieve equality, but this is rarely possible due to the inherent conservativeness of formal guarantees from finite samples. The tightness largely depends on how the model's test accuracy $g(\theta)$ degrades under increasing poisoning. When the degradation is stable and predictable, the learned parameter exhibit more structure, making the NNBC easier to train and the certificate tighter. This highlights a key insight: the regularity of model behavior under poisoning affects not just robustness and accuracy, but also the feasibility of learning a generalizable barrier, reflecting the foundational principle of BCs, which rely on the continuity and predictability of system dynamics in parameter space.

**Influence of number of samples in the data generation process.** The effectiveness of BaRC hinges on its data generation process, which produces ($i$) poisoned training trajectories for learning the NNBC and ($ii$) i.i.d. scenario samples for PAC-style certification. These are controlled by two key parameters: $N$ (number of trajectories) and $\hat{N}$ (number of scenarios). While $\hat{N}$ can be selected based on the desired violation rate $\epsilon$ and confidence level $1 - \beta$ via inequality (17), choosing $N$ is more empirical. Since the NNBC is a learned function, $N$ must be large enough to capture the safe/unsafe boundary, and it depends on the dataset and model complexity. Empirically, BaRC achieves reliable certification when initialized with at least $N = 1000$ samples. If NNBC training or SCP feasibility fails, increasing $N$ by $\sim 500$ typically restores feasibility, balancing statistical coverage with computational cost. This incremental strategy balances cost and coverage, ensuring sufficient data for both learning the NNBC and robust certification.

**Scalability of BaRC.** BaRC is designed to be broadly applicable across architectures, optimizers, and poisoning modalities, and it has demonstrated strong scalability by certifying high-capacity models such as ResNet on CIFAR-10. However, this generality and empirical robustness come at a computational cost since BaRC relies on empirical training trajectories and data generation process, each corresponding to a ML model training run. (see Table 3 for runtime analysis).

**Generality of BaRC.** BaRC models gradient-based training as a discrete-time stochastic dynamical system, operating entirely in parameter space. It assumes no white-box access to the attack (e.g., strategy, trigger, or poisoning ratio), model architecture, loss landscape, or optimizer; all these elements are abstracted into the realized update map $f$, which is observed only through sampled trajectories used to train and verify the barrier certificate $\mathcal{B}_\varphi$. While BaRC directly certifies robustness only against feature-space perturbations (not label corruption), its generality is evident in several ways: ($i$) it certifies both training-time and test-time poisoning, ($ii$) it supports any model architecture and hyperparameters under gradient-based optimizers such as (S)GD or Adam, and ($iii$) it requires no knowledge of the attack strategy or poisoning ratio. Extending BaRC to non-$\ell_p$ perturbations remains a promising direction for future work.

## 7 ETHICS STATEMENT

Our work provides a way to formally assess the worst-case robustness of neural networks against poisoning at both training and test time—addressing them jointly for the first time. Although such insights could, in principle, be misused by adversaries, we argue that identifying and understanding these vulnerabilities is essential for the safe deployment of neural networks now and in the future. We therefore believe that the societal benefits of advancing robustness research outweigh the potential risks, and we do not anticipate any immediate misuse arising from our contributions. In addition, this paper was entirely written by the authors. Large language models (LLMs), were used solely for final-stage language editing and polishing, without contributing to the scientific content or experimental results.

## 8 REPRODUCIBILITY STATEMENT

We have taken considerable care to ensure the reproducibility of our findings. Detailed descriptions of the experimental setup are provided in Sections 5 and D, where we also report all hyperparameter choices. To control stochasticity, we fixed random seeds in all pseudorandom number generators used in the experiments. The complete codebase, along with the configuration files for every experiment, is available at `https://figshare.com/s/42f69e5af3c98213688c` and will be publicly released upon acceptance.

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

## A    NOTATION

We denote the sets of real, positive real, and negative real numbers by $\mathbb{R}$, $\mathbb{R}^+$, and $\mathbb{R}^-$, respectively. The absolute value of a scalar $\theta \in \mathbb{R}$ is denoted by $|x|$. The sets of positive integers and non-negative integers are denoted by $\mathbb{N}$ and $\mathbb{N}_0$, respectively. The set $\mathbb{R}^d$ denotes the $d$-dimensional Euclidean space. We define $[r]$ as the set of the first $r$ natural numbers (i.e., $[r] := \{1, 2, \ldots, r\}$). For any vector $x \in \mathbb{R}^d$, its Euclidean ($\ell_2$-norm) is denoted by $\|x\|_2$, and its infinity norm ($\ell_\infty$-norm) is denoted by $\|x\|_\infty := \max |x_i|$. For any $a \in \mathbb{R}$, the ceiling function $\lceil a \rceil$ returns the smallest integer greater than or equal to a, and the Rectified Linear Unit (ReLU) activation is defined as $\text{ReLU}(a) := \max\{0, a\}$. Finally, for any set $\Theta$, the indicator function $\mathbf{1}_\Theta(\theta)$ equals 1 if $\theta \in \Theta$ and 0 otherwise.

For more clarity, all symbols, notation, and key quantities used throughout the BaRC framework are summarized in Table 2.

Table 2: Summary of key symbols and definitions in the BaRC framework.

| Symbol | Definition / Scope |
|---|---|
| **Threat Model & Perturbation** | |
| $p$ | Norm type for threat model ($\ell_\infty$ in main; $\ell_2$ in Appendix) |
| $\delta, \delta'$ | Max perturbation magnitude (per feature) for train/test |
| $\rho, \rho'$ | Poisoning ratio: fraction of corrupted samples in train/test algorithm |
| $\Delta, \Delta'$ | Feature-space perturbation matrices for train/test data |
| $\mathcal{D}_{\text{train}}^{\Delta}$ | Poisoned training dataset |
| $\mathcal{D}_{\text{test}}^{\Delta'}$ | Poisoned test dataset |
| $\mathbb{P}$ | Distribution over poisoning scenarios for PAC guarantees |
| **Training Dynamics & Parameters** | |
| $\Theta, \Theta_0$ | Parameter space; distribution of initial parameters |
| $\theta(0), \theta(t)$ | Model parameters at initialization and iteration $t$ |
| $t_\infty$ | Terminal time step at training convergence |
| $f(\theta, \mathcal{J})$ | Update rule (e.g., SGD): $\theta - \gamma_t \nabla \mathcal{L}$ |
| $\mathfrak{S}$ | Discrete-time stochastic dynamical system modeling training |
| **Safety criterion and robust radii** | |
| $g(\theta)$ | Accuracy of model $h_\theta$ on test data |
| $g_c(\theta), g_{\text{p}}(\theta)$ | Accuracy on clean vs. poisoned test set |
| $\mathcal{G}(\theta)$ | Test degradation gap: $\mathcal{G} = g_c - g_{\text{p}}$ |
| $\alpha$ | Threshold for maximum allowed test accuracy degradation |
| $\Theta_s, \Theta_u$ | Safe/unsafe sets s.t. $\mathcal{G}(\theta) \leq \alpha / > \alpha$ |
| $\delta_{\text{emp}}, \delta'_{\text{emp}}$ | Empirical robust radius: largest $\delta/\delta'$ such that test accuracy degradation at most $\geq \alpha$ |
| $\delta_{\text{cert}}^*, \delta_{\text{cert}}'^*$ | Certified train/test-time robust radius |
| $g_{\text{p}}^*$ | Certified accuracy ($g_{\text{p}}^* = g_{\text{c}} - \alpha$ as in definition 4) |
| **Neural Barrier Certificate (NNBC)** | |
| $\mathcal{B}_\varphi(\theta)$ | Barrier certificate parameterized by neural weights $\varphi$ |
| $N$ | Number of poisoning trajectories used for NNBC training |
| $\mathcal{B}_\varphi^*$ | Trained NNBC satisfying all loss constraints |
| $\Theta_\mathcal{Z}$ | Feasible domain: sublevel set $\{\theta : \mathcal{B}(\theta) \leq 0\}$ |
| $\mathcal{L}(\varphi)$ | Barrier training loss (Eq. 10) with three ReLU terms |
| $\vartheta$ | Collected dataset for barrier training (initial/safe/unsafe parameters) |
| **Scenario Certification (RCP / SCP)** | |
| $q_1, q_2, q_3$ | Constraint functions encoding the three BC conditions |
| $\eta_r$ | Margin in robust convex problem (RCP) over full scenario space |
| $\eta_s, \eta_s^*$ | Margin in scenario convex problem (SCP) and its optimal value |
| $\hat{N}$ | Number of i.i.d. scenarios used in SCP evaluation |
| $\epsilon$ | Max violation probability allowed over unseen scenarios |
| $\beta$ | Confidence parameter for PAC bound ($1 - \beta$ confidence) |

# B PROOFS

Here we provide the proofs for the results stated in the main part of the paper.

## B.1 PROOF OF THEOREM 10

**Theorem (Certified robust radius).** Let $h_\theta$ be an ML model, trained on a (potentially) poisoned dataset $\mathcal{D}_{\text{train}}^\Delta$ and evaluated on a (potentially) poisoned dataset $\mathcal{D}_{\text{test}}^{\Delta'}$, as in section 2.1. Consider a dt-DS $\mathfrak{S} = (\Theta, \Theta_0, \mathcal{D}_{\text{train}}^\Delta, f)$ as in Definition 6, modeling the training dynamics of $h_\theta$. Let $\delta_{\text{emp}}$ (resp. $\delta'_{\text{emp}}$) denote the empirically derived train-time (resp. test-time) robust radius, and let $\Theta_s \subseteq \Theta$ and $\Theta_u \subseteq \Theta$ denote the corresponding safe and unsafe sets of terminal parameters, as introduced in Definition 4. If there exists a BC $\mathcal{B}$ satisfying the conditions in Definition 7 for a train-time (resp. test-time) robust radius $\delta_{\text{cert}}$ (resp. $\delta'_{\text{cert}}$), then all trajectories initialized at $\theta(0) \in \Theta_0$ remain within the safe set $\Theta_s$ and never enter the unsafe set $\Theta_u$ for any perturbations $\|\Delta\|_p \leq \delta_{\text{cert}}$ (resp. $\|\Delta'\|_p \leq \delta'_{\text{cert}}$). Thus, $\delta_{\text{cert}}$ (resp. $\delta'_{\text{cert}}$) serves as a certified train-time (resp. test-time) robust radius, ensuring that, under worst-case perturbations, the degradation in test accuracy is at most $\alpha$.

*Proof.* By Definitions 4 and 7, the zero-level set of the barrier, $\{\theta \in \mathbb{R}^d \mid \mathcal{B}(\theta) = 0\}$, separates the safe region $\Theta_s := \{\theta \in \mathbb{R}^d \mid \mathcal{G}(\theta) \leq \alpha\}$ from the unsafe region $\Theta_u := \{\theta \in \mathbb{R}^d \mid \mathcal{G}(\theta) > \alpha\}$.
**(1)** Without loss of generality, because the parameters at $t = 0$ are randomly initialized with small magnitudes—hence untrained and uninfluenced by data—the model's predictions are essentially random. Its test accuracy is therefore at chance level, whether evaluated on clean or poisoned inputs. Consequently, the initialization gap satisfies $\mathcal{G}(\theta(0)) \approx 0$, which is negligible relative to any admissible threshold $\alpha$. By Definition 4, it follows that $\theta(0) \in \Theta_s$.
**(2)** By condition (6), the initial model parameters $\theta(0) \in \Theta_0$ always satisfy $\mathcal{B}(\theta(0)) \leq 0$. This aligns with (1). So training begins inside (or on the boundary of) the barrier zero sublevel set.
**(3)** Suppose that at iteration $t$, $\mathcal{B}(\theta(t)) \leq 0$. If $\theta(t) \in \Theta_u$, then condition (7) implies $\mathcal{B}(\theta(t)) > 0$, which is a contradiction. Therefore, $\theta(t)$ must lie in $\Theta_s$.
**(4)** By condition (8), for any admissible poisoning with $\|\Delta\|_p \leq \delta_{\text{cert}}$, if $\mathcal{B}(\theta(t)) \leq 0$, then the next state $\theta(t + 1)$ also satisfies $\mathcal{B}(\theta(t + 1)) \leq 0$ and the zero sub-level set $\{\theta \in \mathbb{R}^d \mid \mathcal{B}(\theta) \leq 0\}$ is forward invariant for the training dynamics under all admissible $\Delta$.
**(5)** From (2)–(4) we conclude that once the training starts inside the safe region, the barrier condition guarantees that $\mathcal{B}(\theta(t)) \leq 0$ holds for every iteration $t$. In particular, at the terminal time $t = t_\infty$ we have $\mathcal{B}(\theta(t_\infty)) \leq 0$, which by the separation property in (3) implies that $\theta(t_\infty) \in \Theta_s$. By Definition 4, this means that the accuracy gap at convergence satisfies $\mathcal{G}(\theta(t_\infty)) \leq \alpha$, that is, the trajectory remains in $\Theta_s$ for all $t$ and never enters $\Theta_u$. For test-time perturbations $\Delta'$, observe that they do not alter the training dynamics and only affect the accuracy at evaluation. Hence, the same separation argument applies: the terminal parameters remain in $\Theta_s$ for all $\Delta'$ with $\|\Delta'\|_p \leq \delta'_{\text{cert}}$.
**(6)** Hence, $\delta_{\text{cert}}$ (resp. $\delta'_{\text{cert}}$) serves as a certified train-time (resp. test-time) robust radius, guaranteeing that under worst-case admissible perturbations the degradation in test accuracy at convergence is bounded by $\alpha$. $\qquad\square$

## B.2 PROOF OF THEOREM 11

**Theorem (BaRC)** Let $h_\theta$ be an ML model, trained on a potentially poisoned training dataset $\mathcal{D}_{\text{train}}^\Delta$ and evaluated on a potentially poisoned test dataset $\mathcal{D}_{\text{test}}^{\Delta'}$, as described in Section 2.1. Assume that model updates follow the gradient-based rule $f$ as in (1), with the training process modeled by a dt-DS $\mathfrak{S}$. Consider $\alpha \in [0, 1]$, $\epsilon \in (0, 1)$, and $\beta \in [0, 1]$ as a gap threshold as in (5), the probability of violation as in (15), and the confidence level as in (17), respectively. Suppose an NNBC $\mathcal{B}_\varphi^*$ is trained using a finite number of samples generated through the procedure in Section 9, producing a certified train-time robust radius $\delta_{\text{cert}}^*$ (resp. certified test-time robust radius $\delta_{\text{cert}}'^*$). Let $\eta_s^* < 0$ be the optimal barrier margin obtained by solving the SCP in (16) on $\hat{N}$ i.i.d. samples, with $\hat{N}$ satisfying the bound in (17). Then, with a confidence of at least $1 - \beta$, the learned certificate $\mathcal{B}_\varphi^*$ ensures that, for all poisoning perturbations satisfying $\|\Delta\|_p \leq \delta_{\text{cert}}^*$ (resp. $\|\Delta'\|_p \leq \delta_{\text{cert}}'^*$), the model's converged parameters remain in the safe set $\mathcal{S}$, and the certified test accuracy $g_b^*(\theta(t_\infty))$ satisfies $\mathcal{G}(\theta(t_\infty)) \leq \alpha$ as in Definition 4, with the violation probability of at most $\epsilon$.

*Proof.* Let $\mathbb{P}$ denote a probability measure on the product space $\Theta \times \mathcal{S}_\Delta$, where $\Theta$ is the parameter space of the ML model and $\mathcal{S}_\Delta$ denotes the set of admissible poisoning perturbation matrices $\Delta$. Our goal is to certify, with high confidence, that the trained model $h_x$ satisfies safety and accuracy constraints even under worst-case poisoning. In the main text, this objective is posed as a robust constrained program (RCP) in (14), where the constraints must hold for all admissible perturbations.

Because solving the RCP is generally intractable—owing to its dependence on the full uncertainty space and the absence of a closed form for $f$—we relax it to a chance-constrained problem (CCP). The CCP permits violations on at most an $\epsilon$ fraction of the uncertainty set, which is acceptable from a probabilistic safety perspective. In this formulation, we define the violation probability $\mathbb{V}(\eta)$ as

$$\mathbb{V}(\eta) := \mathbb{P}\left[(\theta, \Delta) \in \Theta \times \mathcal{S}_\Delta : \exists k \in \{1, 2, 3\} \text{ such that } q_k(\theta, \Delta) > \eta\right]. \tag{18}$$

This quantity is the central object of the CCP, capturing the probability that the BC margin $\eta$ is violated under a random poisoning scenario. We say $\eta$ is $\epsilon$-feasible if $\mathbb{V}(\eta) \leq \epsilon$, i.e., the CCP holds with probability at least $1 - \epsilon$.

To solve the CCP in practice, we approximate it by the SCP in (16), which replaces the probabilistic constraint with empirical constraints over $\hat{N}$ i.i.d. scenarios drawn from $\mathbb{P}$. Concretely, the SCP seeks a margin $\eta$ such that (16) is satisfied.

Let $\eta_s^*$ be the SCP solution constructed from the sample set $\omega = \{(\theta_i, \Delta_i)\}_{i=1}^{\hat{N}} \sim \mathbb{P}$. By the scenario framework of Calafiore & Campi (2006), under standard assumptions (e.g., uniqueness and measurability of the solution), the probability that $\eta_s^*$ violates the original CCP constraint is bounded as

$$\mathbb{P}^{\hat{N}}(\mathbb{V}(\eta_s^*) > \epsilon) \leq \sum_{k=0}^{R-1} \binom{\hat{N}}{k} \epsilon^k (1 - \epsilon)^{\hat{N}-k}, \tag{19}$$

where $\mathbb{P}^{\hat{N}} = \mathbb{P} \times \cdots \times \mathbb{P}$ (taken $\hat{N}$ times) is the product measure on the full multi-sample $\omega$, and $R$ denotes the number of support constraints of the SCP.

In our setting, the trained NNBC $\mathcal{B}_\varphi$ is fixed in (16), and the SCP has a single decision variable (the scalar margin $\eta_s$). Hence the maximal number of support constraints is $R = 1$. Substituting $R = 1$ into the scenario bound yields

$$\mathbb{P}^{\hat{N}}(\mathbb{V}(\eta_s^*) > \epsilon) \leq (1 - \epsilon)^{\hat{N}}. \tag{20}$$

To make this failure probability at most $\beta$, it suffices to require $(1 - \epsilon)^{\hat{N}} \leq \beta$, i.e.

$$\hat{N} \geq \frac{\ln \beta}{\ln(1 - \epsilon)} \quad \left(\text{equivalently, } \hat{N} \geq \left\lceil \frac{\ln \beta}{\ln(1-\epsilon)} \right\rceil \text{ for integer } \hat{N}\right).$$

Therefore, if the number of sampled scenarios $\hat{N}$ satisfies this condition, then with probability at least $1 - \beta$ (over the draw of $\omega$), the SCP solution $\eta_s^*$ is $\epsilon$-feasible for the CCP and thus approximates the RCP by certifying safety and test-accuracy constraints on all but an $\epsilon$-fraction of poisoning scenarios drawn from $\mathbb{P}$.

Finally, if $\eta_s^* < 0$ for some poisoning radius $\delta_{\text{cert}}^*$ (or test-time radius $\delta_{\text{cert}}'^*$), we conclude that, with confidence at least $1 - \beta$, for all poisoning perturbation matrices $\Delta$ satisfying $\|\Delta\|_p \leq \delta_{\text{cert}}^*$ (and analogously for test-time $\Delta'$ with $\|\Delta'\|_p \leq \delta_{\text{cert}}'^*$), the terminal parameters remain in the certified safe set and the test accuracy satisfies $g(\theta(t_\infty)) \geq \alpha$, except on an $\epsilon$-fraction of cases.

This shows how the intractable RCP is relaxed to a CCP and solved via an SCP while preserving formal, probabilistic guarantees on robust generalization; the BaRC framework inherits these guarantees through this layered connection, completing the proof.

$\square$

# C ALGORITHMS

We summarize the certification procedure in Figure 4, which illustrates the overall workflow. In addition, Algorithms 2 and 3 describe the certification process under train-time and test-time poisoning settings, respectively. These procedures detail how the NNBC is trained and verified using disjoint parameter sets to provide valid robustness guarantees.

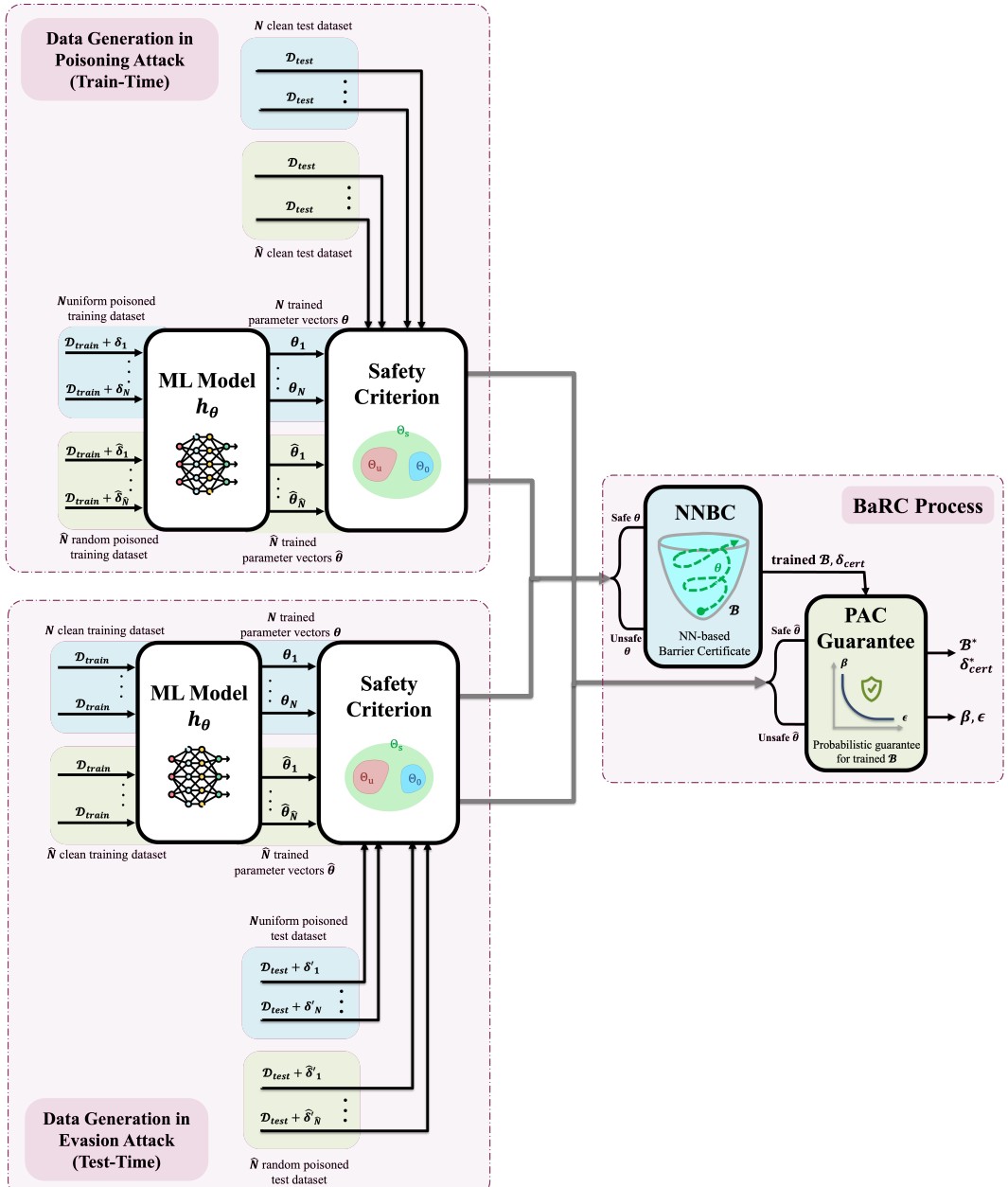

Figure 4: **BaRC framework.** The left panel illustrates the data generation process under both train-time poisoning attacks and test-time evasion attacks. For each perturbation level, the model $h_\theta$ is trained on perturbed datasets to produce two disjoint sets of parameter vectors: $\theta$ and $\hat{\theta}$. A safety criterion function is then applied to each parameter vector to label it as safe or unsafe. The set $\theta$ is used to train an NNBC $\mathcal{B}_\varphi$, while the set $\hat{\theta}$ is used to evaluate $\mathcal{B}_\varphi$ through a scenario-based PAC analysis. The BaRC process (right panel) outputs a certified NNBC $\mathcal{B}_\phi^*$, and its corresponding robustness radius $\delta_{\text{cert}}^*$ or $\delta_{\text{cert}}'^*$, and a probabilistic guarantee with violation probability at most $\epsilon$ and a confidence of at least $1-\beta$.

---

**Algorithm 2** BaRC for an ML model $h_\theta$ Against Data Poisoning Attack

---

**Input** Clean train and test datasets $\mathcal{D}_{\text{train}}$ and $\mathcal{D}_{\text{test}}$, model $h_\theta$, gap threshold $\alpha$, training horizon $t_\infty$, norm $p$, number of samples for NNBC $N$, number of samples for SCP $\hat{N}$, step size $d_\delta$, confidence level $1 - \beta \in [0, 1]$, poison ratio $\rho \in [0, 1]$, max iterations $T$;
**Output** Certified radius $\delta^*_{\text{cert}}$, NNBC $\mathcal{B}^*_\varphi$, violation probability $\epsilon$;

---

**Step 1 - Data Generation for BarC**

---

1: **for** $i = 1$ to $N$ **do**
2:     Initialize $\theta_i(0) \in \Theta_0$ and add to $\mathcal{I}$;
3:     Sample poisoning level $\delta_i$;
4:     Create $\rho$-ratio poisoned dataset $\mathcal{D}^{\Delta_i}_{\text{train}}$ with $\|\Delta_i\|_p = \delta_i$;
5:     **for** $j = 1$ to $t_\infty$ **do**
6:         Train $h_\theta$ on $\mathcal{D}^{\Delta_i}_{\text{train}}$ to obtain $\theta_i(j)$;
7:     **end for**
8:     Evaluate test accuracy $\mathcal{G}(\theta_i(t_\infty))$ on $\mathcal{D}_{\text{test}}$ as in definition 4;
9:     **if** $\mathcal{G}(\theta_i(t_\infty)) \geq \alpha$ **then**
10:         Add $\theta_i(t_\infty)$ to $\mathcal{S}$;
11:     **else**
12:         Add $\theta_i(t_\infty)$ to $\mathcal{U}$;
13:     **end if**
14: **end for**
15: Set $\vartheta \leftarrow \mathcal{I} \cup \mathcal{U} \cup \mathcal{S}$;
16: Compute $\delta_{\text{emp}} \leftarrow \max \{\delta_i \mid g(\theta_i(t_\infty)) \geq \alpha\}$;
17: **return** $\mathcal{I}, \mathcal{U}, \vartheta, \delta_{\text{emp}}$;

---

**Step 2 - BaRC Process**

---

18: Generate $N$ trajectories to form $\mathcal{I}, \mathcal{U}, \vartheta$ along with corresponding empirical robust radius $\delta_{\text{emp}}$
    and generate $\hat{N}$ i.i.d. trajectories to form $\mathcal{Z}_1, \mathcal{Z}_2, \vartheta'$ from Step 1;
19: Initialize $\delta_{\text{cert}} \leftarrow \delta_{\text{emp}}$, NNBC $\mathcal{B}_\varphi$, and counter $k \leftarrow 0$;
20: **while** $\delta_{\text{cert}} > 0$ **do**
21:     **while** $\mathcal{L} \neq 0$ and $k < T$ **do**
22:         Train $\mathcal{B}_\varphi$ using $\mathcal{I}, \mathcal{U}, \vartheta$ with loss $\mathcal{L}$ as in (10);
23:         Update $\mathcal{L}$ and increment $k \leftarrow k + 1$;
24:         **if** $\mathcal{L} \neq 0$ and $k \geq T$ **then**
25:             Decrease radius: $\delta_{\text{cert}} \leftarrow \delta_{\text{cert}} - d_\delta$ **break**;
26:         **else if** $\mathcal{L} = 0$ **then**
27:             Solve SCP as in (16) using $\mathcal{Z}_1, \mathcal{Z}_2, \vartheta'$ to obtain margin $\eta^*_s$;
28:             **if** $\eta^*_s > 0$ **then**
29:                 Decrease radius: $\delta_{\text{cert}} \leftarrow \delta_{\text{cert}} - d_\delta$;
30:                 (Optional: Increase $N$ );
31:             **else**
32:                 $\delta^*_{\text{cert}} \leftarrow \delta_{\text{cert}}$;
33:                 $\mathcal{B}^*_\varphi \leftarrow \mathcal{B}_\varphi$;
34:                 Compute $\epsilon$ from $\hat{N} = \left\lceil \frac{\ln(\beta)}{\ln(1-\epsilon)} \right\rceil$;
35:                 **break**
36:             **end if**
37:         **end if**
38:     **end while**
39: **end while**
40: **return** $\mathcal{B}^*_\varphi, \delta^*_{\text{cert}}, \epsilon$

---

---

**Algorithm 3** BaRC for an ML model $h_\theta$ Against Evasion Attack

---

**Input** Clean train and test datasets $\mathcal{D}_{\text{train}}$ and $\mathcal{D}_{\text{test}}$, model $h_\theta$, gap threshold $\alpha$, training horizon $t_\infty$, norm $p$, number of samples for NNBC $N$, number of samples for SCP $\hat{N}$, step size $d_\delta$, confidence level $1 - \beta \in [0, 1]$, poison ratio $\rho' \in [0, 1]$, max iterations $T$;
**Output** Certified radius $\delta'^*_{\text{cert}}$, NNBC $\mathcal{B}^*_\varphi$, violation probability $\epsilon$;

---

### Step 1 - Data Generation for BarC

---

1: **for** $i = 1$ to $N$ **do**
2:     Initialize $\theta_i(0) \in \Theta_0$ and add to $\mathcal{I}$;
3:     **for** $j = 1$ to $t_\infty$ **do**
4:         Train $h_\theta$ on $\mathcal{D}_{\text{train}}$ to obtain $\theta_i(j)$;
5:     **end for**
6:     Sample poisoning level $\delta'_i$;
7:     Create $\rho$-ratio poisoned dataset $\mathcal{D}_{\text{test}}^{\Delta'_i}$ with $\|\Delta'_i\|_p = \delta'_i$;
8:     Evaluate test accuracy $\mathcal{G}(\theta_i(t_\infty))$ on $\mathcal{D}_{\text{test}}^{\Delta'_i}$ as in definition 4;
9:     **if** $\mathcal{G}(\theta_i(t_\infty)) \geq \alpha$ **then**
10:         Add $\theta_i(t_\infty)$ to $\mathcal{S}$;
11:     **else**
12:         Add $\theta_i(t_\infty)$ to $\mathcal{U}$;
13:     **end if**
14: **end for**
15: Set $\vartheta \leftarrow \mathcal{I} \cup \mathcal{U} \cup \mathcal{S}$;
16: Compute $\delta'_{\text{emp}} \leftarrow \max \{\delta'_i \mid g(\theta_i(t_\infty)) \geq \alpha\}$;
17: **return** $\mathcal{I}, \mathcal{U}, \vartheta, \delta'_{\text{emp}}$

---

### Step 2 - BaRC Process

---

18: Generate $N$ trajectories to form $\mathcal{I}, \mathcal{U}, \vartheta$ along with corresponding empirical robust radius $\delta_{\text{emp}}$
    and generate $\hat{N}$ i.i.d. trajectories to form $\mathcal{Z}_1, \mathcal{Z}_2, \vartheta'$ from Step 1;
19: Initialize $\delta'_{\text{cert}} \leftarrow \delta'_{\text{emp}}$, NNBC $\mathcal{B}_\varphi$, and counter $k \leftarrow 0$;
20: **while** $\delta'_{\text{cert}} > 0$ **do**
21:     **while** $\mathcal{L} \neq 0$ and $k < T$ **do**
22:         Train $\mathcal{B}_\varphi$ using $\mathcal{I}, \mathcal{U}, \vartheta$ with loss $\mathcal{L}$ as in (10);
23:         Update $\mathcal{L}$ and increment $k \leftarrow k + 1$;
24:         **if** $\mathcal{L} \neq 0$ and $k \geq T$ **then**
25:             Decrease radius: $\delta'_{\text{cert}} \leftarrow \delta'_{\text{cert}} - d_\delta$; **break**
26:         **else if** $\mathcal{L} = 0$ **then**
27:             Solve SCP as in (16) using $\mathcal{Z}_1, \mathcal{Z}_2, \vartheta'$ to obtain margin $\eta^*_s$;
28:             **if** $\eta^*_s > 0$ **then**
29:                 Decrease radius: $\delta'_{\text{cert}} \leftarrow \delta'_{\text{cert}} - d_\delta$;
30:                 (Optional: Increase $N$ );
31:             **else**
32:                 $\delta'^*_{\text{cert}} \leftarrow \delta'_{\text{cert}}$;
33:                 $\mathcal{B}^*_\varphi \leftarrow \mathcal{B}_\varphi$;
34:                 Compute $\epsilon$ from $\hat{N} = \left\lceil \frac{\ln(\beta)}{\ln(1-\epsilon)} \right\rceil$;
35:                 **break**
36:             **end if**
37:         **end if**
38:     **end while**
39: **end while**
40: **return** $\mathcal{B}^*_\varphi, \delta'^*_{\text{cert}}, \epsilon$

---

# D  ADDITIONAL EXPERIMENTS

**Additional results.**  Figure 5 plots certified accuracy $g_{\mathrm{p}}^*$ versus the perturbation radius $\delta$ across attacks, models, and datasets, for both train- and test-time settings. In most configurations, the certified robust radius $\delta_{\mathrm{cert}}^*$ closely tracks the empirical radius $\delta_{\mathrm{emp}}^*$ (especially when accuracy degrades smoothly) highlighting the tightness of BaRC's certificates in those regimes. In addition, a quantitative comparison with RAB under clean-label backdoor attacks (Table 5) shows that BaRC achieves similarly strong certified radii and more tightly aligns with the empirical robustness. NNBC $\mathcal{B}_\varphi$ are trained with Adam using the multi-term loss in equaion (10). To balance penalties across constraint sets, we normalize the weights by set size: $c_\mathcal{I} = \frac{1}{N_\mathcal{I}}$, $c_\mathcal{U} = \frac{1}{N_\mathcal{U}}$, $c_\mathcal{Z} = \frac{1}{N_\mathcal{Z}}$, where $N_\mathcal{I}, N_\mathcal{U}, N_\mathcal{Z}$ are the cardinalities of the initial, unsafe, and feasible sublevel sets, respectively. This compensates for variations induced by the choice of the gap threshold $\alpha$.

**Configurations.**  All experiments were implemented in PyTorch (Python 3.11) and run on two environments: (A) a MacBook Pro with Apple M3 Pro (12-core CPU), 36 GB RAM, macOS Sonoma 14.4; (B) 4× NVIDIA H100 GPUs with a 16-core CPU, and 64 GB RAM. Hardware configurations are denoted abstractly as **A** and **B** in the tables. Full experimental settings appear in Table 3, with corresponding figures in the last column; model architectures are listed in Table 4.

Table 3: Experimental configurations across datasets, attacks, optimizers, models, and NNBC settings. The **Attack** block lists the poisoning method (PGD, BPA, BDA, AA), perturbation norm ($p$-norm), poisoning ratio, and step size. **ML Setup** specifies the baseline model, optimizer, and learning rate used for training. The **Certificate** block highlights how robustness guarantees are constructed: a neural barrier certificate (NNBC) is learned directly from the parameters of the trained ML model and then validated via a **PAC** bound, which provides a formal generalization guarantee on unseen data by bounding the violation probability $\epsilon$ (confidence 99.99%). **Execution Setup** records hardware abstraction (HW) and runtime (minutes).

| Dataset | Attack | | | | ML Setup | | Certificate | | | | | | | Execution Setup | | |
|---|---|---|---|---|---|---|---|---|---|---|---|---|---|---|---|---|
| | Type | $p$-norm | $\rho$ | Step | Model | Optimizer ($l_r$) | Type | NNBC | | | PAC | | HW | Run time | | Fig. |
| | | | | | | | | $N$ | Layer | Optimizer ($l_r$) | $\tilde{N}$ | $\epsilon$ | | (min) | | |
| MNIST | PGD | $\infty$ | 1 | 40 | CNN | SGD (0.01) | Train-Time | 4000 | 5 | Adam (0.001) | 1800 | 0.005 | A | 53 | | 5a |
| MNIST | BPA | $\infty$ | 0.3 | 30 | MLP | GD (0.10) | Train-Time | 5000 | 7 | Adam (0.001) | 2500 | 0.003 | A | 36 | | 5d |
| MNIST | PGD | 2 | 1 | 40 | CNN | Adam (0.001) | Train-Time | 3000 | 5 | Adam (0.01) | 1500 | 0.006 | A | 19 | | 5g |
| MNIST | PGD | $\infty$ | 1 | 40 | MLP | SGD (0.01) | Train-Time | 4000 | 5 | Adam (0.001) | 2000 | 0.004 | A | 31 | | 5h |
| MNIST | BDA | $\infty$ | 0.1 | 40 | CNN | SGD (0.01) | Test-Time | 3000 | 5 | Adam (0.001) | 1500 | 0.006 | A | 18 | | 5k |
| MNIST | AA | 2 | 1 | 100 | MLP | Adam (0.001) | Test-Time | 4000 | 7 | Adam (0.10) | 2500 | 0.003 | A | 26 | | 5n |
| SVHN | PGD | $\infty$ | 0.9 | 40 | CNN | Adam (0.001) | Train-Time | 2000 | 4 | Adam (0.001) | 800 | 0.011 | A | 39 | | 5b |
| SVHN | BDA | $\infty$ | 0.1 | 30 | MLP | GD (0.10) | Train-Time | 4000 | 5 | Adam (0.001) | 1500 | 0.006 | B | 13 | | 5e |
| SVHN | PGD | 2 | 0.9 | 30 | MLP | SGD (0.01) | Train-Time | 4000 | 5 | Adam (0.001) | 1500 | 0.006 | B | 27 | | 5f |
| SVHN | PGD | 2 | 0.9 | 40 | CNN | SGD (0.01) | Train-Time | 2000 | 4 | Adam (0.001) | 800 | 0.011 | A | 73 | | 5i |
| SVHN | BDA | 2 | 0.9 | 40 | CNN | SGD (0.10) | Test-Time | 2500 | 4 | Adam (0.001) | 600 | 0.015 | A | 25 | | 5j |
| SVHN | AA | 2 | 0.8 | 100 | CNN | SGD (0.01) | Test-Time | 3000 | 4 | Adam (0.001) | 1000 | 0.009 | A | 81 | | 5m |
| SVHN | AA | 2 | 0.8 | 100 | MLP | GD (0.10) | Test-Time | 4000 | 5 | Adam (0.001) | 2000 | 0.004 | A | 63 | | 5o |
| SVHN | BPA | $\infty$ | 0.2 | 30 | ResNet | Adam (0.10) | Train-Time | 4000 | 4 | Adam (0.001) | 1000 | 0.009 | B | 72 | | 5p |
| SVHN | PGD | 2 | 0.9 | 40 | ResNet | Adam (0.01) | Train-Time | 3000 | 5 | Adam (0.001) | 700 | 0.015 | B | 84 | | 5s |
| CIFAR10 | PGD | $\infty$ | 0.8 | 40 | CNN | Adam (0.001) | Train-Time | 1500 | 4 | Adam (0.001) | 200 | 0.045 | A | 138 | | 5c |
| CIFAR10 | BDA | $\infty$ | 0.1 | 40 | CNN | Adam (0.10) | Test-Time | 1500 | 4 | Adam (0.001) | 200 | 0.045 | B | 66 | | 5l |
| CIFAR10 | BPA | 2 | 0.2 | 30 | ResNet | Adam (0.01) | Train-Time | 2000 | 4 | Adam (0.001) | 600 | 0.015 | B | 81 | | 5q |
| CIFAR10 | BDA | $\infty$ | 0.2 | 100 | ResNet | Adam (0.10) | Train-Time | 3000 | 4 | Adam (0.001) | 600 | 0.015 | B | 76 | | 5r |
| CIFAR10 | AA | 2 | 0.3 | 30 | ResNet | Adam (0.01) | Test-Time | 2000 | 4 | Adam (0.001) | 400 | 0.022 | B | 98 | | 5t |

Table 4: Architectural specifications of models used across datasets. **Conv Layers** reports the number of convolutional layers and their output channels. **Pooling** specifies the type and frequency of downsampling. **FC Layers** denotes the fully connected layers with hidden dimensions up to the output layer. **Params (M)** provides the approximate number of trainable parameters (in millions).

| Dataset | Model | Conv Layers | Pooling | FC Layers | Params (M) |
|---|---|---|---|---|---|
| MNIST | CNN | 3 conv (32, 64, 128) | 3× MaxPool (2×2) | 3 FC (256, 128, 10) | ~1.2M |
| | MLP | – | – | 4 FC (512, 256, 128, 10) | ~0.6M |
| | LeNet | 2 conv (16, 32) | 2× AvgPool (2×2) | 3 FC (240, 120, 10) | ~0.1M |
| SVHN | CNN | 2 conv (64, 128) | 2× MaxPool (2×2) | 3 FC (256, 128, 10) | ~1.5M |
| | MLP | – | – | 4 FC (1024, 512, 256, 10) | ~3.2M |
| | ResNet18 | 18 conv (standard) | Global AvgPool | 1 FC (10) | ~11M |
| CIFAR-10 | CNN | 3 conv (64, 128, 256) | 3× MaxPool (2×2) | 3 FC (512, 256, 10) | ~4.5M |
| | ResNet18 | 18 conv (standard) | Global AvgPool | 1 FC (10) | ~11M |

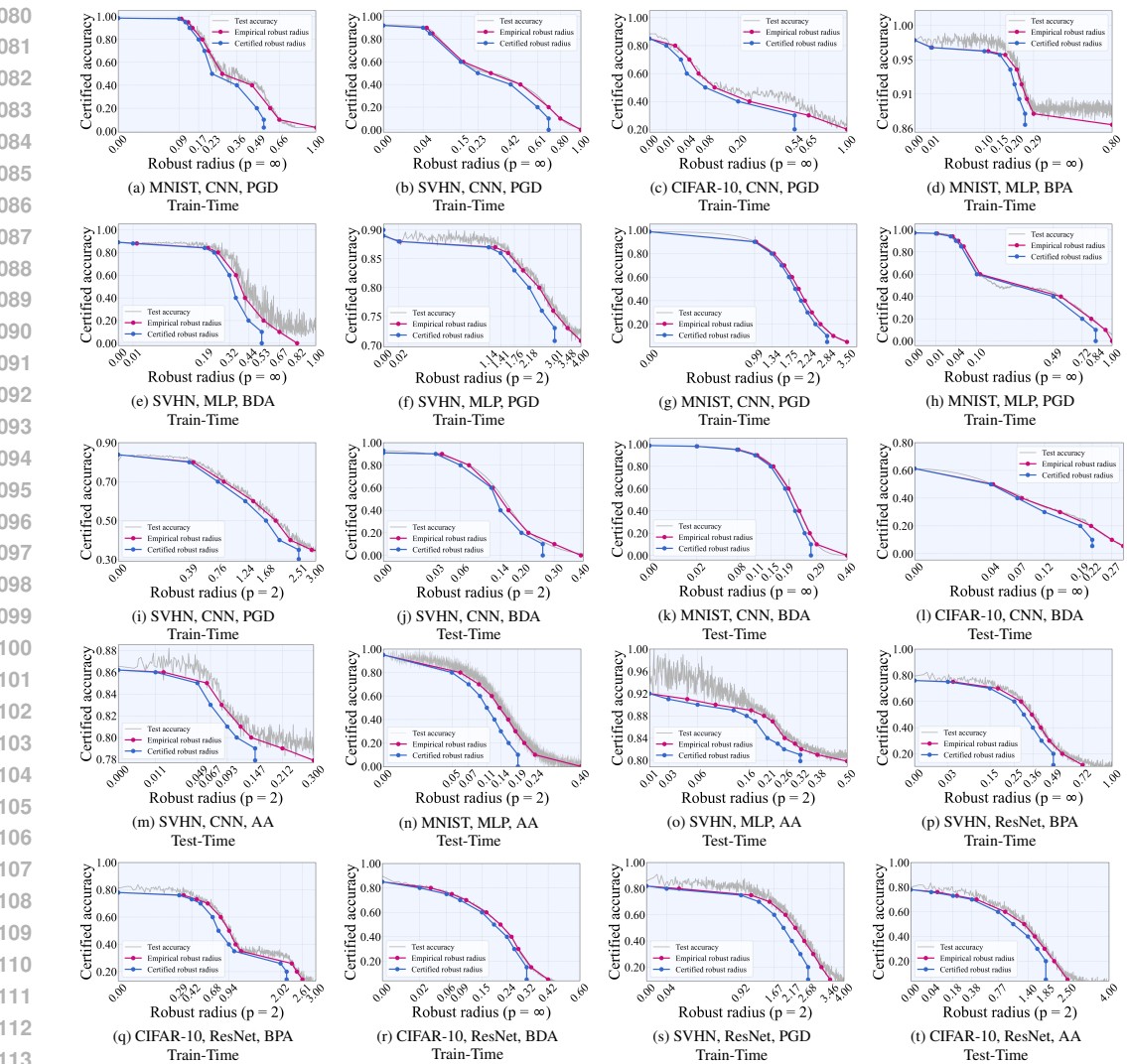

Figure 5: Certified accuracy versus perturbation magnitude $\delta$ under different poisoning scenarios and datasets. Each subplot shows the test accuracy $g$, empirical robust radius $\delta_{\text{emp}}$, and certified robust radius $\delta^*_{\text{cert}}$ under the proposed BaRC framework. The confidence level is fixed at $1 - \beta$ with $\beta = 10^{-4}$. Violation probabilities are: $\epsilon =$ (a) 0.005, (b) 0.011, (c) 0.045, (d) 0.003, (e) 0.006, (f) 0.006, (g) 0.006, (h) 0.004, (i) 0.011, (j) 0.015, (k) 0.006, (l) 0.045, (m) 0.009, (n) 0.003, (o) 0.004, (p) 0.009, (q) 0.015, (r) 0.015, (s) 0.015, (t) 0.022.

| Dataset | Optimizer | $\mathcal{A}$ | $\rho$-ratio | $p$-norm | $g^*_{\text{p}}$ | $\delta_{\text{emp}}$ | $\delta^*_{\text{cert}}$ (RAB) | $\delta^*_{\text{cert}}$ (BaRC) |
|---------|-----------|---------------|--------------|----------|------------------|-----------------------|-------------------------------|--------------------------------|
| MNIST | SGD | BDA | 0.15 | $\infty$ | 0.90 | 0.08 | NA | 0.08 |
|  |  |  |  |  | 0.80 | 0.16 | 0.10 | 0.15 |
|  |  |  |  |  | 0.60 | 0.22 | 0.14 | 0.19 |
| SVHN | SGD | BDA | 0.1 | 2 | 0.80 | 0.09 | NA | 0.06 |
|  |  |  |  |  | 0.60 | 0.12 | 0.07 | 0.11 |
|  |  |  |  |  | 0.40 | 0.16 | 0.08 | 0.14 |
| CIFAR-10 | Adam | BDA | 0.1 | $\infty$ | 0.50 | 0.05 | NA | 0.04 |
|  |  |  |  |  | 0.40 | 0.09 | NA | 0.07 |
|  |  |  |  |  | 0.30 | 0.15 | 0.05 | 0.12 |

Table 5: Comparison of certified robust radii obtained by **BaRC** and **RAB** under identical poisoning settings and CNN architecture. While RAB is the only directly comparable baseline available, it is limited to test-time certification and supports only a narrow class of attacks, specifically, **Backdoor Attacks (BDA)**. In contrast, BaRC has no such restriction and produces certified radii that closely match empirical robustness, even in cases where RAB fails to certify the target test accuracy.

# E   ADDITIONAL DISCUSSION

**How BaRC is fully agnostic?**   BaRC models gradient-based training as a dt-SD, and reasons purely in parameter space. The certificate uses only $(i)$ observed parameter trajectories $\{\theta(t)\}$ from training runs under admissible perturbations and $(ii)$ a terminal safety predicate $\mathcal{G}(\theta) \leq \alpha$. It assumes no white-box access to the attack (strategy, trigger, poisoning ratio), the network architecture, the loss landscape, or the optimizer/scheduler; all of these are subsumed into the realized update map $f$ and are reflected only through the trajectories (the empirical reachable set) on which the BC $\mathcal{B}(\theta)$ is trained and verified. Consequently, the same construction applies to both train-time and test-time perturbations (only the terminal labels change) and to arbitrary model classes and training pipelines without threat-model tuning. As qualitative support, Figure 6 contrasts training trajectories of 20,000 parameters of the *same* CNN on clean SVHN (left) yields compact, near-stationary trajectories, whereas a BDA-poisoned SVHN (right) exhibits early drift and dispersion. Configuration details are immaterial, the point is that poisoning reshapes the empirical reachable set, and BaRC's barrier exploits this separability to carve a safe sublevel set that retains clean runs while excluding poisoned ones.

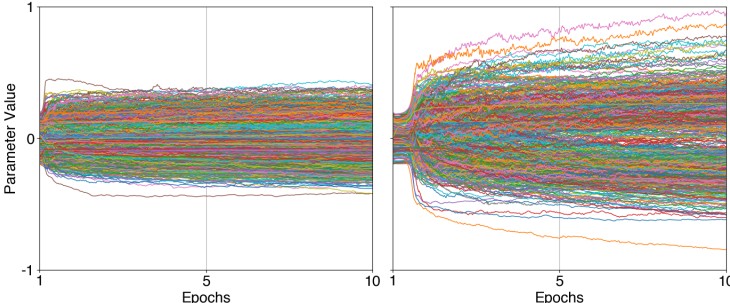

Figure 6: Training trajectories of 20,000 parameters from a CNN on SVHN. **Left:** CNN trained on clean data. **Right:** The same CNN trained on SVHN with BDA poisoning. The clear visual separation between the two regimes illustrates the core intuition behind BaRC's framework.

**If the model is initially safe, why do we still need to train it?**   In BaRC, an initial parameter assumed as safe state because it lies inside the certified region, but this is not a judgment of model utility. At initialization the network is untrained; any apparent accuracy can be incidental to a particular split and is not a reliable indicator of quality. The purpose of training the ML model is to optimize the learning objective on the training data, drive the empirical loss to a target level, and satisfy a clear convergence or stopping criterion (for example, a gradient-norm threshold or a fixed training horizon). Until this optimization occurs, discussing accuracy is largely uninformative, since the training loss remains high even if it satisfies the safety criterion. The barrier certificate reconciles safe initialization with the need to learn. Being safe at the start only authorizes us to begin from an allowed region. The forward-invariance condition ensures that, while we minimize the loss, the optimization trajectory remains within the safe set and the terminal parameters meet the prescribed robustness tolerance. In short, safe-at-start does not equal trained-or-useful; training is indispensable for reducing loss and achieving acceptable performance, and BaRC guarantees that this learning process remains within certified safety at convergence.

**Gap vs. certified accuracy.**   Based on definition 4, we can fix the certified accuracy under poisoning as well, that is, choose the minimum test accuracy level we want to guarantee for $g_{\mathrm{p}}(\theta(t_\infty))$ and find the corresponding robust radius. This is exactly equivalent to fixing the gap threshold $\alpha$, because the relation $\mathcal{G}(\theta) = g_c(\theta) - g_{\mathrm{p}}(\theta)$ is linear. Once the clean accuracy at the terminal model $g_c(\theta(t_\infty))$ is a number, setting a target for $g_{\mathrm{p}}(\theta(t_\infty))$ is the same as setting $\alpha$ via $\alpha = g_c(\theta(t_\infty)) - g_{\mathrm{p}}^*(\theta(t_\infty))$, and conversely a fixed $\alpha$ immediately implies the certified $g_{\mathrm{p}}(\theta(t_\infty)) = g_c(\theta(t_\infty)) - \alpha$. We report certified accuracy for readability, but reporting the gap is interchangeable; both lead to the same safe/unsafe split and the same certified radius.

**Challenging settings sometimes yield tighter certificates!**
We occasionally observe that more challenging configurations (harder datasets or larger architectures) produce certified radii $\delta_{\text{cert}}^*$ that track the empirical radii $\delta_{\text{emp}}^*$ more tightly. The primary driver is not the dataset difficulty per se, but the stronger models these settings necessitate (e.g., ResNet in place of MLP/CNN). Such architectures typically achieve higher clean accuracy and, more importantly for certification, induce a smoother degradation of test accuracy as the perturbation radius $\delta$ increases. Because our barrier separates safe from unsafe based on this accuracy–radius curve, smoother trajectories in parameter space and more regular accuracy decline make

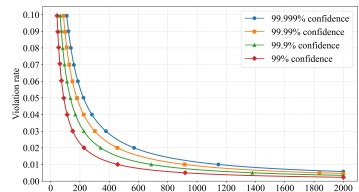

Figure 7: Violation rate $\epsilon$ vs the number of scenarios $\hat{N}$ used for solving the SCP at different confidence level.

the safe set easier to approximate, yielding tighter certificates. That benefit comes with a statistical trade-off in our PAC check. Harder settings are computationally heavier. Since the PAC bound scales with $\hat{N}$, a smaller $\hat{N}$ directly increases the certified violation probability $\epsilon$ (See Figure 7). Consequently, on configurations like CIFAR-10 with a ResNet, one may see a tight $\delta_{\text{cert}}^*$ (thanks to the smoother accuracy–radius behavior of the stronger model) but a higher $\epsilon$ than in easier settings. In short: stronger models can improve tightness of the certified radius, while computational constraints in challenging regimes can worsen the PAC violation rate.

**On seemingly extreme poisoning ratios.** In several figures we intentionally show very large corruption ratios (e.g., $0.5 - 1$). This is by design and reflects what BaRC certifies: a bound on the perturbation magnitude per sample (the robust radius) without assuming any fixed fraction of corrupted points. The fraction (how many samples the adversary touches) is treated as unknown and can range anywhere in $[0, 1]$; the statement of our certificate does not include this ratio. Why, then, display large ratios? Two reasons: $(i)$ to stress, test the method with challenging scenarios and illustrate that the guarantee is decoupled from the corruption fraction; and $(ii)$ to provide a clear contrast with prior works that typically fix the ratio and certify how many points can be corrupted. BaRC makes the opposite design choice: we place a hard limit on the size of admissible perturbations and certify robustness regardless of how many points the attacker modifies.

**Role of RCP, CCP, SCP, and PAC.** Achieving robustness certification for all possible poisoning trajectories is unfeasible due to the requirement to address an infinite number of perturbation scenarios and to know the closed form of the map $f$. BaRC navigates this complexity by employing the RCP $\Rightarrow$ CCP $\Rightarrow$ SCP $\Rightarrow$ PAC approach: it avoids comprehensive safety verification (RCP) by tolerating a minor probability of violation (CCP), verify the validity of the certificate in a limited set of scenarios (SCP), and apply PAC bounds to ensure that the learned barrier is broadly applicable with some level of confidence. This method allows BaRC to resolve a complex robustness issue via a manageable data-centric method with assured formal guarantees derived from statistical learning theory.

**Effect of margin $\eta_s^*$.** The SCP margin $\eta_s^*$ measures how well the trained barrier certificate $\mathcal{B}_\vartheta$ satisfies the safety constraints over the sampled scenarios. Specifically, $\eta_s^* < 0$ implies that all constraints are strictly satisfied, confirming that the NNBC is valid under the given certified radius and the scenarios sampled. Thus, the more negative the margin, the more robust the barrier appears against the sampled violations. If $\eta_s^* > 0$, the barrier does not satisfy at least one constraint, implying that the NNBC must be retrained or the certified radius must be reduced. Once $\eta_s^* \leq 0$, the learned certificate is accepted with confidence $1 - \beta$ and the probability of violation at most $\epsilon$, according to the PAC guarantee.

**Empty sets and robust radius adjustment.** The empirical labeling of models as safe or unsafe, defined in equation (9), depends on a fixed threshold $\alpha \in [0, 1]$ applied to the test accuracy $g(\theta_i(t_\infty))$. If $\alpha$ exceeds the clean accuracy at $\delta = 0$, all models are labeled unsafe and $\mathcal{S} = \emptyset$; if it falls below the worst-case accuracy at $\delta = \delta_{\max}$, all are labeled safe and $\mathcal{U} = \emptyset$. These degenerate cases invalidate the empirical margin $\eta_s^*$ required by the SCP. To ensure feasibility, we adopt the following convention: if $\mathcal{S} = \emptyset$, we set the certified robust radius $\delta_{\text{cert}}$ (or $\delta_{\text{cert}}'$ for test-time) to zero. If $\mathcal{U} = \emptyset$, we increase $\alpha$ until the dataset becomes non-empty and SCP verification succeeds. The smallest such threshold is then used as the effective certified $\alpha$. This explains a recurring pattern

in our evaluation plots, such as Figure 2, where the final certified radius often equals the previous one despite being computed at a lower $\alpha$. In these cases, certification at the lower threshold fails (either due to infeasibility or lack of data) and we conservatively reuse the last valid radius to avoid overstating robustness.

**Effect of sampling density on robust radius curves.** In certain cases, the test-accuracy curve appears to drop below the robust-radius curves, which should not occur from a theoretical standpoint. This discrepancy arises from insufficient sampling of $\alpha$ values when computing $\delta_{\mathrm{cert}}$, leading to interpolation errors. The fidelity of both certified and empirical robust-radius curves depends on the density of these evaluation points: increasing the number of sampled $\alpha$ thresholds produces curves that more accurately reflect the true robustness profile. In Figure 8, we increase the sampling density from 10 $\alpha$ points in (a) to 20 in (b). The resulting curves in (b) are smoother and more accurate, and no longer exhibit anomalous crossing. This confirms that denser sampling yields more precise and theoretically consistent robust-radius estimates in both empirical and certified settings.

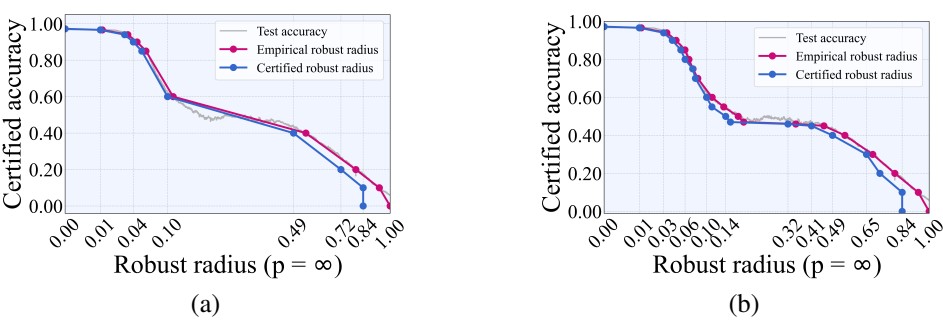

Figure 8: MNIST, MLP, PGD, Train-Time

**Why we report attack parameters despite being attack-agnostic?** Although BaRC is inherently *attack-agnostic*, we still present details such as attack types, poisoning ratios, datasets, and model architectures. These specifications are not required by the framework itself, which certifies robustness solely from training trajectories, independent of adversarial strategy, corruption level, or model family. Rather, they are reported to provide clarity and to illustrate the scope and strength of BaRC when subjected to a wide range of adversarial conditions.

**On the limitations of trajectory separability** Our certification framework implicitly relies on the assumption that training trajectories under clean and poisoned data remain sufficiently separable in parameter space, allowing the NNBC to distinguish safe from unsafe regions. One might ask whether an advanced adversary could engineer a poisoning strategy whose trajectory stays arbitrarily close to the clean one, thereby making the learning of a separating certificate infeasible. Indeed, in cases where the perturbation is extremely weak, the resulting degradation in test accuracy is negligible, so that clean and poisoned trajectories are effectively indistinguishable. In such regimes, however, the model is not meaningfully threatened: the attack has little practical impact, and a robustness certificate is not required in the first place. Thus, the difficulty of separating nearly identical trajectories is directly aligned with the lack of adversarial effect, reinforcing that BaRC is most relevant in regimes where poisoning causes non-trivial accuracy degradation.

**Related Work.** While there has been limited progress in computing a certified $\ell_p$-norm poisoning radius for a desired model accuracy, the general literature on formal certificates for data poisoning robustness remains even more nascent. Different data poisoning certification approaches include: ($i$) **Ensemble-based methods** that partition the training dataset and train base classifiers *independently*. A final ensemble classifier aggregates its predictions (e.g., via majority voting, run-off election), and robustness is certified by analyzing the clean sample majority needed to withstand poisoning (Levine & Feizi, 2021; Jia et al., 2021; Cohen et al., 2019; Wang et al., 2022; Rezaei et al., 2023). These methods generally assume independence among base classifiers and allow *unbounded perturbation budgets*. ($ii$) **Randomized Smoothing** is a technique inspired by test-time probabilistic certificates (Cohen et al., 2019). Several works adapt this idea by introducing randomness into the training process. These approaches guarantee robustness by assuming a *fixed bounded perturbation* through

averaging model behavior over perturbed training datasets to certify label corruption (Rosenfeld et al., 2020), specific backdoor patterns (Weber et al., 2023; Wang et al., 2020), and both data feature and label corruptions (Zhang et al., 2022). $(iii)$ **Differential Privacy** based approaches leverage theoretical connections between privacy and robustness to certify the models (Ma et al., 2019; Xie et al., 2023). $(iv)$ **Model-specific methods** usually assume *bounded perturbation budgets* and a *bounded number of poisoned samples*. For instance, in the case of graph neural networks, Gosch et al. (2025) leverages the kernel equivalence of neural networks and develops mixed integer linear programming-based certificates using graph neural tangents (Sabanayagam et al., 2023) for $\ell_p$ norm based feature corruption. (Sabanayagam et al., 2025) extends the framework developed in Gosch et al. (2025) to label corruptions. Sosnin et al. (2025) proposes a gradient-based certification method for neural networks using convex overapproximations of parameter trajectories. However, their relaxations tend to be loose, especially as the training progresses, and their method is tightly coupled to specific architectures, making generalization to broader model classes difficult.

In contrast to prior work, BaRC provides the first general-purpose framework for certifying the robust radius in the case of poisoning based on training dynamics. By modeling training as a discrete-time dynamical system and leveraging barrier certificates from control theory, BaRC enables formal certification across both train- and test-time poisoning settings. Crucially, BaRC does not require model-specific assumptions, adversary knowledge, or white-box access, and can be applied to a wide class of models trained via (stochastic) gradient descent optimization, including Adam or momentum-based methods.

