# OpenReview forum: "BaRC: Barrier Robustness Certificates for Neural Networks against Data Poisoning and Evasion Attacks"
_ICLR.cc/2026/Conference — ICLR 2026 Conference Withdrawn Submission_

### Official Review · Reviewer_UVCp · 2025-10-16

**Soundness:** 2
**Presentation:** 3
**Contribution:** 1
**Rating:** 2
**Confidence:** 3

**Summary:**

The paper presents a principled approach to a formal robustness certification framework. It formulates typical training as a discrete-time dynamical system, and then recasts poisoning robustness as a safety verification problem within this framework. Barrier certificates for this system are approximated as a neural network, and from that a probably approximately correct bound is obtained. The certification covers both training and test time poisoning and is agnostic to the underlying model being certified.

**Strengths:**

The paper is well written and presented. The motivation to unify train and test time poisoning consideration is clearly articulated and the formal approach is reasonably clearly laid out (minor points noted below) with a well-developed train of argument.

**Weaknesses:**

The main issue is that I am not confident that the results presented are a useful or reliable bound. The approach, if I understand it correctly, is essentially basing the hypothesised bound on a dataset of empirical bounds, evaluated from a known attack. As such, there is no confidence that the bound could not be broken by an as-yet unidentified attack approach/class with effectiveness significantly outside of the attacks used for deriving this bound.

Further to the above, the fact that the claimed bound is "tight", i.e. close to the empirical values, is surely just a reflection of the fact that it is derived from them, rather than anything inherently tight about it. Similarly, the claim in section 6 that the bound derivation requires "no knowledge of the attack strategy" is questionable.

The above could seem to explain why the more traditional randomised smoothing bound they compare with in Figure 3, i.e. the RS bound is conservative as it has to cover all possible known and as-yet unknown attacks.  (It is also worth noting that RS bounds are typically on a different (smoothed) classifier and not the original one. So this is also perhaps not comparing apples with apples)

The extensive training runs required make this approach very computationally expensive. That said, it is an upfront cost, c.f. Randomised Smoothing, which imposes a performance cost at run-time.

In section 3, there is no discussion of how the dataset is perturbed - randomly vs adversarial attack for example. Just that some perturbation size is defined. Further, there does not seem to be any guidance in how to define the values in the grids \delta_grid

**Questions:**

- The concerns expressed above re the bound being very closely tied to the attack method chosen for the underlying poisoned training samples could be explored by looking at how the certified bound varies when the attack algorithm is changed (while keeping the model, optimiser, etc fixed). If it does not change substantially, then this would support the method. But if it tracks with the attack effectiveness, confidence would be reduced in the approach.

- In section 3, the NNBC seems to be trained just based on terminal theta values, i.e. per equation (9). Has the trajectory information been thrown away, and could it be used more effectively?

- Why in section 3 is the approach confined to either training or test poisoning. Surely the benefit of the combined framework is that both can be considered together?

- In section 2.1, is the map l:YxY to R+ a map from the labels or from some set of softmax values? With the former, as it seems to be defined, it seems to be less amenable to the gradient approach used for training

- In sec 2.2, there is a statement that at test-time, evasion attacks can shift the decision boundary. Surely that boundary is fixed in training. Not sure what is meant here.

- In Definition 4, is it stated that g_c is trained on clean and g_p trained on poisoned. Both are evaluated at the same theta in the definition of G. So in cases where the training set is not poisoned (but the test set is), G would be identically zero? I would imagine you mean to evaluate one on the perturbed test set?

---

### Official Review · Reviewer_Zxe2 · 2025-10-27

**Soundness:** 3
**Presentation:** 3
**Contribution:** 3
**Rating:** 4
**Confidence:** 4

**Summary:**

The paper proposes BaRC, a framework to certifying the robustness of neural networks against training-time data poisoning and test-time data evasion attacks. The method tracts the model training as the dynamical system with the discrete time and applies barrier certificates to verify that the training trajectory in the parameter space belongs to a "safe" region under norm-bounded adversarial perturbations.

**Strengths:**

The authors formulate the problem of certification against data poisoning and data evasion attacks as the verification in the parameter space and applies BC to build theoretically grounded framework of certification against norm-bounded adversarial perturbations both during training and during inference. The paper is well-written, the method is model-agnostic and the idea is elegant (to me). The experiments are mostly convincing.

**Weaknesses:**

There are several weaknesses to discuss during rebuttal. I believe the authors can comment on them; I am willing to increase my score.

1) The proposed NNBC network trains on a finite number of sampled trajectories. If the undermining boundary (the set of $\theta$ corresponding to $\mathcal{B}(\theta) = 0$) is complex or poorly estimated by samples, the resulting certificate may be overly conservative. It would be nice to have an algorithm to estimate the approximation error in a realistic scenario (for example, where no i.i.d. assumption on trajectories is made, since it may not hold in adversarial settings).

2) The set of attacks considered is narrow: only norm-bounded perturbations are within the scope of work. How can BaRC be extended to support label flipping attacks or non-additive perturbations?

3) The set of competitors to compare BaRC against is narrow; more recent related works could be added, for example [1], [2].

[1] Philip Sosnin, Mark Niklas Muller, Maximilian Baader, Calvin Tsay, and Matthew Wicker. Certified ¨
robustness to data poisoning in gradient-based training.
[2] Lukas Gosch, Mahalakshmi Sabanayagam, Debarghya Ghoshdastidar, and Stephan Gunnemann. ¨
Provable robustness of (graph) neural networks against data poisoning and backdoor attacks.
Transactions on Machine Learning Research, 2025.

**Questions:**

See weaknesses.

---

### Official Review · Reviewer_KGoo · 2025-10-30

**Soundness:** 1
**Presentation:** 2
**Contribution:** 2
**Rating:** 2
**Confidence:** 3

**Summary:**

The authors present a barrier style certification, that attempts to demonstrate robustness to both poisoning and evasion attacks.

**Strengths:**

On the surface this is an interesting idea in an active area. And, on the whole, it's well written in a general sense, although I did note multiple cases (see below) where a lack of specificity hamstrung the authors ability to convey their point.

**Weaknesses:**

While I called out the quality of the writing in the strengths, I will note that the narrative (distinct from the technical quality of the writing) read a little imbalanced at times. For example, the introduction and abstract are almost exclusively focused upon poisoning, before quickly also diverting into evasion attacks? And, fundamentally, why is the evasion attack occurring on a population level (with the fraction of samples) - what is the motivation here from both the attacker and defender?

More broadly, I do not feel that the authors have made a clear case about why evasion and poisoning attacks need to be considered in concert. For example, what would stop one from employing certifications against poisoning and evasion attacks? And if there is a joint certification, how would this joint certification compare to individual evasion / poisoning style certifications? To me, this would be an interesting point for experimental evaluation, one that is missing within the work.

More broadly, when it comes to the actual mechanics of the work, I also have a concern about the neural network barrier certificate. If I am understanding the paper correctly, I'm left concerned about what circumstances would one be looking for a certification, whose value is approximated, and potentially vulnerable to adversarial manipulation?   Unless I'm misunderstanding things, the authors then calculate a 99.99% confidence interval on the approximation - but to me, any reasonable reader who is not familiar with the details of this would treat this as a true 99.99% confidence interval. Which underscores the problem - a technique that appears to have rigor, but is built upon an approximation, has the potential to give rise to a false sense of confidence in vulnerable users. I will very much acknowledge that I may have misunderstood this point, but I am intrinsically highly cautious about any works that would even suggest such an approach.

Finally, before I go into more minor comments, I'm left utterly unconvinced by the authors argument about the scalability of their approach.  Line 474 states "has demonstrated strong scalability by certifying high-capacity models such as ResNet on CIFAR-10". Being able to scale to CIFAR-10 is no longer a claim that holds any credulity at all, base CIFAR-10 calculations can be performed on consumer hardware from 10 years ago. Such experiments are not within community expectations for robustness papers. Also as an aside, Resnet models are presented as ResNet-18, to denote the actual size of model used.

There are also some overstated claims of novelty (see L122 below).

The following notes were compiled partially to justify my position, but also to provide suggestions to the authors on areas that they may wish to improve:
- L15: "Most existing defenses lack formal guarantees or rely on restrictive assumptions about the model class, attack type, extent of poisoning, or point-wise certification" - most implies that defences are all of once class, rather than being normal and certifiable classes. The "or point-wise certification" part is also a non-sequitur here.
- L23: "To make this practical," the subject of the this is ambiguous - it could be trajectories, norm based data poisoning, barrier certs or more.
- L70: "The approaches are limited to specific architectures in some cases" - placing the qualifier "in some cases" at the end of the statement makes the statement read as being more significant than it actually is. By highlighting this point, the authors present robustness as if it is primarily hamstrung by this issue, yet there are techniques (like randomised smoothing style approaches) which make no assumptions regarding the architecture.
- L122: "Weber et al. (2023) extends randomized smoothing, a test-time certification strategy, to poisoning" - Webber isn't the only work to do this, but this framing implies that they're unique. "Enhancing the Antidote: Improved Pointwise Certifications against Poisoning Attacks" and "Multi-level certified defence against poisoning attacks in offline reinforcement learning" by Liu et. al. are two that I know of off the top of my head.
- L157: "or evaluate an ML model hθ may be adversarially perturbed, resulting in degraded performance. Such poisoning attacks" technically if it's at evaluation time it's not a poisoning attack. Performance is also generally an aggregate metric, whereas evaluation time manipulations of data are often specific.
- L157-161: Discussion about \ell_p is ambiguous about if it's sample-wise, or set wise. Eqn 3 makes it clear that it's a maximal bound on the perturbations across each sample in the set of poisoned samples.
- L199 "and converges at t = t_inf" - doesn't grammatically work in the context fo the rest of the sentence.
- L200-204: Seems to be primarily structured around poisoning, rather than evasion attacks.
- L215: There's not a clear definition of what a trajectory is to this point, nor what an admissible budget realization is.
- L228 "as in [D]efinition 4".

**Questions:**

1: Under what threat model does an \ell_p norm bound matter for data poisoning? Or, to put this another way, under what conditions would an attacker be limited to an \ell_p data perturbation, if they have the ability to corrupt the input data set?

2: Why is theta defined at t in N_0, yet there needs to be an assumption about convergnece at t = t_inf?

3: Does the accuracy drop (L201) rely upon the idea that accuracy must be a metric? Also this relies upon an additive model of accuracy drop - would there be any reason why a multiplicative delta might be more appropriate?

4: "the learning rate \gamma_t is treated as a fixed hyperparameter" - fixed in what sense? The notation would suggest it's varying with time?

5: It may have been a long few weeks, with too many papers to review, but could the authors explain Figure 2. You have a test accuracy plotted as a certified accuracy, against varying robust radii?

6: How does this approach scale? The tested datasets and models are small.

---

### Official Review · Reviewer_nuNw · 2025-11-10

**Soundness:** 3
**Presentation:** 3
**Contribution:** 3
**Rating:** 6
**Confidence:** 3

**Summary:**

This paper proposes Barrier Robustness Certificates, which employ a functions to separate the safe and unsafe regions in parameter space, to derive probabilistic robustness guarantees for both backdoor attacks and evasion attacks. The training process is modelled by a discrete-time dynamic system and the barrier certificate functions are parameterized by neural networks trained on empirical observations. The certified accuracy is then probabilistic depending on the number of empirical samplings.

**Strengths:**

++ It is novel to model robust learning as a dynamical system and utilise the corresponding tools to derive certified robustness.

++ The theoretical bounds, despite probabilistic, are guaranteed for general cases. The methods to obtain this probabilistic bounded are implementable.

++ The method and the theorem are generic in nature, applicable to both training-time and test-time robustness, various model architectures and datasets.

**Weaknesses:**

1. I am not sure about the computational complexity of the proposed method in Algorithm 1, because it include sampling $\hat{N}$ instances and iterates among different value of perturbations $\delta_{cert}$. I would be better to have a discussion about computational complexity of the proposed algorithm.

2. Equation (15) is not the strict guarantee of robustness, making the result of Algorithm 1 weaker than strict robustness guarantee, such as the ones for evasion attacks. The sampling can only bridge the gap between (16) and (15) instead of between (15) and (14).

3. The experiment part is generally good, but the reported results lack some baselines. The authors can considering using some certified robustness against evasion attacks (like IBP-augmented training and randomised smoothing) to compare the (probabilistic) bounded between these methods and the the one derived by the theorem in this manuscript.

4. The author claim that the certification is for the whole dataset instead of instance test data. However, the evaluation function $g$ depends on the test set given. Therefore, I think the conclusion in this manuscript still depends on an empirical test set, which is no much difference from the instancewise certification.

Overall, I am not an expert in this field, but I think the work in this manuscript made some meaningful contributions to the community.

**Questions:**

Please answer the questions raised in the weakness part. In addition, I have the following question:

1. Does the theoretical analyses consider the stochastic optimization in training models? We typically use mini-batch training in which the gradient has a variance. Will this affect the theoretical results?

---

### Note · Authors · 2025-11-14

I have read and agree with the venue's withdrawal policy on behalf of myself and my co-authors.